# Protein Kinase D2 drives chylomicron-mediated lipid transport in the intestine and promotes obesity

Jonathan Trujillo-Viera[1,†] (ID), Rabih El-Merahbi[1,†] (ID), Vanessa Schmidt[1], Till Karwen[1], Angel Loza-Valdes[2] (ID), Akim Strohmeyer[3,4,5], Saskia Reuter[1], Minhee Noh[1] (ID), Magdalena Wit[2] (ID), Izabela Hawro[2] (ID), Sabine Mocek[3,4,5], Christina Fey[6], Alexander E Mayer[1], Mona C Löffler[1], Ilka Wilhelmi[7,8], Marco Metzger[6], Eri Ishikawa[9,10], Sho Yamasaki[9,10], Monika Rau[11], Andreas Geier[11], Mohammed Hankir[12] (ID), Florian Seyfried[12], Martin Klingenspor[3,4,5] (ID) & Grzegorz Sumara[1,2,*] (ID)

## Abstract

**Lipids are the most energy-dense components of the diet, and their overconsumption promotes obesity and diabetes. Dietary fat content has been linked to the lipid processing activity by the intestine and its overall capacity to absorb triglycerides (TG). However, the signaling cascades driving intestinal lipid absorption in response to elevated dietary fat are largely unknown. Here, we describe an unexpected role of the protein kinase D2 (PKD2) in lipid homeostasis. We demonstrate that PKD2 activity promotes chylomicron-mediated TG transfer in enterocytes. PKD2 increases chylomicron size to enhance the TG secretion on the basolateral side of the mouse and human enterocytes, which is associated with decreased abundance of APOA4. PKD2 activation in intestine also correlates positively with circulating TG in obese human patients. Importantly, deletion, inactivation, or inhibition of PKD2 ameliorates high-fat diet-induced obesity and diabetes and improves gut microbiota profile in mice. Taken together, our findings suggest that PKD2 represents a key signaling node promoting dietary fat absorption and may serve as an attractive target for the treatment of obesity.**

**Keywords** chylomicron; fat absorption; intestine; obesity; protein kinase D2/PKD2/PRKD2

**Subject Categories** Digestive System; Metabolism

## Introduction

The type of diet plays a major role in modulating organismal metabolism. Diets containing elevated fat content are generally more energy dense, which promotes a positive energy balance and, consequently, obesity. The digestive system is the first site to be challenged by elevated levels of fat in the diet. After emulsification of ingested fat by bile acids, triglycerides are broken down into glycerol, monoglyceride, and fatty acids (FAs) by pancreatic lipases in the small intestine lumen (Lowe, 2002; Hussain, 2014). Monoglycerides and FAs are then taken up by enterocytes either by passive diffusion or by an active mechanism involving FA transporters such as cluster of differentiation 36 (CD36) (Xu *et al*, 2013; Hussain, 2014). In enterocytes, FAs and monoglycerides or glycerol are re-esterified at the endoplasmic reticulum (ER). These monoglyceride and glycerol 3-phosphate pathways are responsible for the majority of TG synthesis in enterocytes (Yang & Nickels, 2015). Following re-esterification, TG are packed into pre-chylomicrons together with lipoproteins such as apolipoprotein B48 (APOB48) and apolipoprotein A4 (APOA4) by the microsomal transfer protein (MTTP)

1   Rudolf-Virchow-Zentrum, Center for Integrative and Translational Bioimaging, University of Würzburg, Würzburg, Germany
2   Nencki Institute of Experimental Biology, Polish Academy of Sciences, Warszawa, Poland
3   Chair for Molecular Nutritional Medicine, Technical University of Munich, TUM School of Life Sciences Weihenstephan, Freising, Germany
4   EKFZ - Else Kröner-Fresenius-Center for Nutritional Medicine, Technical University of Munich, Munich, Germany
5   ZIEL - Institute for Food & Health, Technical University of Munich, Freising, Germany
6   Fraunhofer Institute for Silicate Research (ISC), Translational Center Regenerative Therapies (TLC-RT), Würzburg, Germany
7   Department of Experimental Diabetology, German Institute of Human Nutrition Potsdam-Rehbruecke, Nuthetal, Germany
8   German Center for Diabetes Research (DZD), München-Neuherberg, Germany
9   Molecular Immunology, Research Institute for Microbial Diseases (RIMD), Osaka University, Suita, Japan
10  Molecular Immunology, Immunology Frontier Research Center (IFReC), Osaka University, Suita, Japan
11  Division of Hepatology, University Hospital Würzburg, Würzburg, Germany
12  Department of General, Visceral, Transplant, Vascular and Pediatric Surgery, University Hospital Würzburg, Würzburg, Germany
   *Corresponding author. Tel: +49 931 31 89263 or +48 22 5892 190; E-mails: grzegorz.sumara@uni-wuerzburg.de or g.sumara@nencki.edu.pl
   †These authors contributed equally to this work.

(Mansbach & Siddiqi, 2016). APOB48 is absolutely required for pre-chylomicron formation at the ER (Mansbach & Siddiqi, 2016), while APOA4 is likely responsible for determining final chylomicron size (Lu *et al*, 2006; Kohan *et al*, 2012; Weinberg *et al*, 2012; Kohan *et al*, 2015). Following their assembly, pre-chylomicrons are then transported to the Golgi apparatus to undergo further chemical modifications (possibly including lipidation) and, finally, are designated for secretion (Hesse *et al*, 2013).

Increased dietary fat content leads to the elevation in expression and activity of enzymes critical for lipid uptake, FA re-esterification, TG packing, and lipoproteins required for assembly of pre-chylomicrons (Petit *et al*, 2007; Hernández Vallejo *et al*, 2009; Clara *et al*, 2017). Interestingly, an increase in chylomicron size might be a major factor determining the elevated capacity of enterocytes to process excessive dietary fat (Uchida *et al*, 2012). However, the signaling cascades driving the adaptation of enterocytes to increased lipid loads in the intestinal lumen remain largely unknown.

Protein kinase D (PKD) family members are diacylglycerol (DAG) and protein kinase C (PKC) effectors, which recently emerged as central regulators of nutrient homeostasis (Sumara *et al*, 2009; Löffler *et al*, 2018; Mayer *et al*, 2019; Kolczynska *et al*, 2020). The PKD family includes three members (PKD1, PKD2, and PKD3), which regulate several aspects of cellular metabolism and pathophysiology (Fielitz *et al*, 2008; Kim *et al*, 2008; Sumara *et al*, 2009; Kleger *et al*, 2011; Konopatskaya *et al*, 2011; Rozengurt, 2011; Ittner *et al*, 2012; Löffler *et al*, 2018; Mayer *et al*, 2019; Kolczynska *et al*, 2020; preprint: Mayer *et al*, 2020). Our recent study suggested that PKDs might be activated in response to free FAs (FFAs) or DAG (Mayer *et al*, 2019). Moreover, high-fat diet (HFD) feeding activated PKDs in the liver (Mayer *et al*, 2019). However, the impact of PKDs activity on lipid metabolism in the intestine has not been investigated so far.

Here, we show that PKD2 (also known as PRKD2) is activated upon lipids loading in intestine and promotes chylomicron growth and lipidation and consequently TG secretion by human and mouse enterocytes. Interestingly, PKD2 directly phosphorylates one of the apolipoproteins associated with chylomicrons, namely APOA4. Deletion of PKD2 in intestine of mice or in human enterocytes results in increased abundance of intracellular and secreted APOA4. Consistently with these results, the ablation of PKD2 activity or the specific deletion of this kinase in the intestine resulted in reduced absorption of fat, increased excretion of energy in the feces and resistance to high-fat diet-induced obesity. Moreover, deletion of PKD2 resulted in resistance to high-fat diet-induced diabetes and pathological changes in the gut microbiota. Additionally, we demonstrate that a PKD-specific inhibitor decreases fat absorption and is effective in the treatment of obesity and associated diseases in animal models. Finally, our data indicate that human subject activity of PKD2 in the intestine correlates with TG levels in blood. Therefore, we establish PKD2 as a key component of the intestinal fat absorption and an attractive target for future anti-obesity therapies.

## Results

### PKD2 inactivation protects from diet-induced obesity

Our previous studies revealed that PKD1 promotes obesity by blocking energy dissipation in adipocytes (Löffler *et al*, 2018), while PKD3 promotes hepatic insulin resistance (Mayer *et al*, 2019). However, the role of PKD2 in regulating glucose and lipid metabolism and in the development of obesity-induced diabetes is not known. We addressed this by utilizing mice with global defective PKD2 enzymatic activity, because of point mutations in serines 707 and 711 to alanines ($Pkd2^{ki/ki}$ mice) (Matthews *et al*, 2010a). We maintained $Pkd2^{ki/ki}$ mice and corresponding control animals ($Pkd2^{wt/wt}$) on a normal chow diet (ND) or high-fat diet (HFD) for 22 weeks after weaning. Remarkably, while $Pkd2^{ki/ki}$ and $Pkd2^{wt/wt}$ mice on ND gained similar weight, $Pkd2^{ki/ki}$ mice maintained on HFD gained significantly less weight than corresponding control animals (Fig 1A). Weight reduction in $Pkd2^{ki/ki}$ mice fed HFD was associated with decreased adiposity and reduced adipocyte size (Fig 1B–E). However, the weight of other organs was not affected by PKD2 inactivation (Fig 1C). As previous studies indicated that deletion of PKD1 kinase (closely related to PKD2), in adipose tissue, promotes the expression of beige adipocyte-specific markers (Löffler *et al*, 2018), we have tested the expression of *Ucp1, Cidea, Bmp7, Prdm16, Ppara, Pgc1a, Adrb3, Cidec, Myh2, Ckm, Mck, Slc6a8, Slc27a2, Ucp3,* and *Myh1* subcutaneous adipose tissue of $Pkd2^{ki/ki}$ and control $Pkd2^{wt/wt}$ mice fed HFD and except *Slc6a8*, which was downregulated in the mice without active PKD2; there were no significant changes in the expression of these genes (Fig EV1A). Similarly, liver histology and markers of hepatic function, aspartate transaminase (AST), and alanine transaminase (ALT) did not differ between genotypes, while hepatic content of TG was decreased in

**Figure 1. PKD2 inactivation protects from diet-induced obesity.**

A Body weight gain of male mice with the specified genotypes under normal (ND) or high-fat diet (HFD).

B Quantification of fat, free fluid, and lean mass by nuclear magnetic resonance (NMR) of mice in HFD in panel (A).

C Organ weight (percentage of total weight) of different fat depots, liver, and quadriceps of $Pkd2^{wt/wt}$ and $Pkd2^{ki/ki}$ male mice after 22 weeks in HFD.

D Quantification of the average adipocyte size in SubWAT and EpiWAT of male mice of the specified genotypes in normal and high-fat diet.

E Representative pictures of H&E staining of SubWAT and EpiWAT of indicated male mice fed HFD.

F Triglyceride content in liver of $Pkd2^{wt/wt}$ and $Pkd2^{ki/ki}$ male mice in panel (A). Relative to $Pkd2^{wt/wt}$.

G Glucose tolerance test of the specified genotypes after 16 weeks in HFD.

H Insulin tolerance test of $Pkd2^{wt/wt}$ and $Pkd2^{ki/ki}$ male mice after 18 weeks in HFD.

I, J Triglycerides (I) and free fatty acids (J) in circulation of specified mice after 18 weeks in HFD.

K, L Energy expenditure (K) and energy intake (L) of mice after 20 weeks in HFD.

Data information: Male mice were subjected to ND or HFD directly after weaning, and the specified metabolic parameters were measured. $n = 7$ for ND. In HFD, $n = 7$ for $Pkd2^{wt/wt}$ and $n = 8$ for $Pkd2^{ki/ki}$. Data presented as average ± SEM. *$P \leq 0.05$, **$P \leq 0.01$, ***$P \leq 0.001$. Significances were assessed by using a two-tailed Student's *t*-test for independent groups.

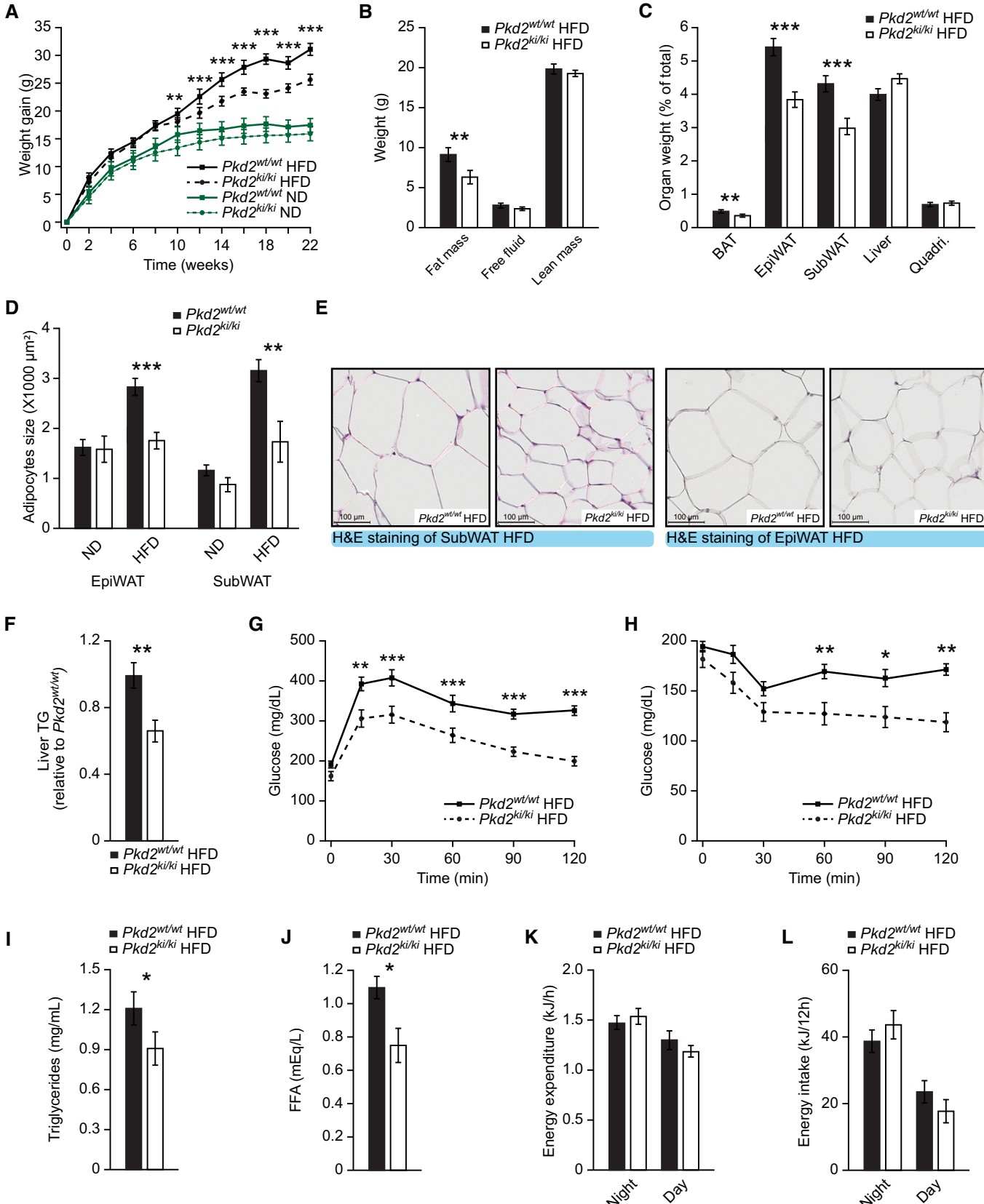

Figure 1.

$Pkd2^{ki/ki}$ mice (Figs 1F and EV1B–D). Of note, mice deficient for PKD2 enzymatic activity were protected from diet-induced glucose intolerance and displayed better insulin sensitivity when fed HFD (Fig 1G and H). Moreover, in the $Pkd2^{ki/ki}$ mice, we found a small but not significant increase in insulin levels during the glucose-stimulated insulin secretion test (Fig EV1E). Furthermore, the islet area (relative to pancreas area) is increased in $Pkd2^{ki/ki}$ mice (Fig EV1F–H). Inactivation of PKD2 in mice fed HFD also reduced circulating triglyceride and free FA levels (Fig 1I and J). Altogether, these findings indicate that inactivation of PKD2 protects from diet-induced obesity as well as associated hyperglycemia and hyperlipidemia.

In order to investigate the mechanisms underlying the amelioration of obesity in mice deficient for PKD2 activity, we used integrated analyses of metabolic parameters which revealed that inactivation of PKD2 does not affect food intake, energy expenditure, or voluntary movements of mice fed HFD (Figs 1K and L, and EV1I), suggesting that reduced body weight gain of $Pkd2^{ki/ki}$ upon HFD feeding must be caused by misregulation of other processes.

## PKD2 promotes the absorption of lipids from food

Since inactivation of PKD2 in mice did not affect food intake and energy expenditure, we hypothesized that absorption of nutrients might be reduced in the absence of PKD2 activity. Therefore, we collected feces from $Pkd2^{ki/ki}$ mice and control animals. We observed that feces collected from $Pkd2^{ki/ki}$ mice fed HFD were yellowish in contrast to the excrements derived from $Pkd2^{wt/wt}$ control mice which displayed a typical dark-brown color (Fig 2A). Moreover, in contrast to stool from control mice, feces from $Pkd2^{ki/ki}$ mice did not sink in water (Fig 2A). Additionally, $Pkd2^{ki/ki}$ mice fed HFD produced more stool than control animals per week also when extrapolated to food intake (Fig 2B and C). Moreover, HFD feeding was significantly less efficient in promoting body weight gain in mice expressing inactive PKD2 relative to control animals (Fig 2D). Of note, $Pkd2^{ki/ki}$ fed ND produced the same amount of feces as corresponding control animals and their color did not differ from feces from control animals (Fig EV2A and B). These data suggest that PKD2 inactivation dramatically modulates the physicochemical properties of feces of mice fed HFD but not of mice fed ND.

Subsequently, we analyzed the energy deposition and chemical composition of stool from mice of both genotypes. Feces derived from $Pkd2^{ki/ki}$ mice were more energy dense and contained more lipids than excrements from $Pkd2^{wt/wt}$ animals (Fig 2E and F). Consequently, $Pkd2^{ki/ki}$ mice excreted more energy than corresponding control animals (Fig 2G), which represent a significant decrease in the total metabolizable energy (Fig 2H). Altogether, these findings suggest that PKD2 promotes obesity by increasing the capacity of the organism to absorb fat from the calories-dense HFD.

## PKD2 promotes the release of TG from enterocytes

To directly test whether PKD2 promotes TG absorption in the intestine, at first we have tested if PKD2 is activated upon ingestion of TG. In fact, we have observed an increase in PKD2 activity in animals that received olive oil gavage after short-term fasting (Fig 3A). Next, we subjected $Pkd2^{ki/ki}$ and $Pkd2^{wt/wt}$ to HFD for 1 week, and after overnight fasting, we orally administered a defined dose of olive oil. Olive oil challenge evoked over an order of magnitude increase in circulating TG levels in control animals, while the same dose of olive oil resulted in significantly lower TG levels in $Pkd2^{ki/ki}$ mice at all the time-points (Fig 3B). To test whether the decreased levels of TG in $Pkd2^{ki/ki}$ mice result from increased uptake of TG in peripheral organs or decrease absorption in the intestine, we have blocked the action of lipoprotein lipase by injection of tyloxapol to inhibit TG uptake by the cells from circulation. Importantly, upon ingestion of olive oil, $Pkd2^{ki/ki}$ mice treated with tyloxapol still presented decreased TG levels in the circulation (Fig 3C). These data strongly suggest that PKD2 activity in the intestine promotes TG absorption.

Upon ingestion of lipid-containing food, fat is first emulsified by bile acids, and then, pancreatic lipase, with the help of colipase, digests TG into FFAs and monoglycerides, which can subsequently be taken up by enterocytes (Lowe, 2002). The inactivation of PKD2 in mice did not affect the expression of pancreatic lipase or colipase in the pancreas as well as protein levels of pancreatic lipase in the duodenum (Fig EV2C and D). Similarly, the expression of genes determining bile acid metabolism, transport, and biosynthesis in the small intestine (ileum) and liver was not affected in mice deficient for PKD2 activity (Fig EV2E). Moreover, the expression of FFA transporters in the intestine was not affected by the inactivation of PKD2 (Fig EV2F). We found it noteworthy that the inactivation of PKD2 in the intestine did not affect the expression of PKD1 and PKD3 (Fig EV2G). Moreover, the intestinal protein abundance of PKD1, PKD2, and PKD3 was not affected by the inactivation of PKD2 (Fig EV2H–K). Interestingly, as confirmed by two antibodies specific for PKD2, on the Western blot a band corresponding to PKD2 appears at a higher molecular weight than the one specific for PKD1 in intestine and in Caco2 cells depleted from PKD2 by shRNA (Fig EV2H and J). Similarly, the signal specific for phosphorylated serine 876 in PKD2 (pPKD2 S876—an active form of PKD2) appears at the same molecular weight as the one corresponding for PKD2 and it was not present in the extracts from the intestine isolated from $Pkd2^{ki/ki}$ mice (Fig 3D). We have also utilized an antibody which recognizes the phosphorylation (activation) of PKD1 (S916) and PKD2 (S876); while the lower band corresponding to the pPKD1 was not altered by inactivation of PKD2, the upper band was present in the extracts from intestine isolated from control mice but not from $Pkd2^{ki/ki}$ mice (Fig 3D). This was true also for Caco2 cells depleted from PKD2 (Fig 3E). All of these largely exclude the possibility of any indirect function of other PKD family members in the intestine as a compensatory mechanism.

Malabsorption of lipids might also be caused by developmental defects of the intestine often manifested by decreased corrugations of the intestinal epithelium, altered cellular composition, or deregulated proliferation of stem cells located in the crypt (Mazzawi et al, 2015; König et al, 2016). Histological analyses revealed that the inactivation of PKD2 did not influence the morphology of the intestine (Fig EV3A). To assess the impact of PKD2 on intestinal organogenesis, we developed organoids derived from stem cells of the crypts isolated from $Pkd2^{ki/ki}$ and $Pkd2^{wt/wt}$ mice (Zietek et al, 2015). PKD2 inactivation did not affect organoid development, size, or cellular composition as revealed by staining for markers of gastrointestinal endocrine cells (chromogranin A) and epithelial brush border regulator (Villin) (Figs 3F and EV3B and C). Finally, paracellular permeability of the intestinal lumen might affect

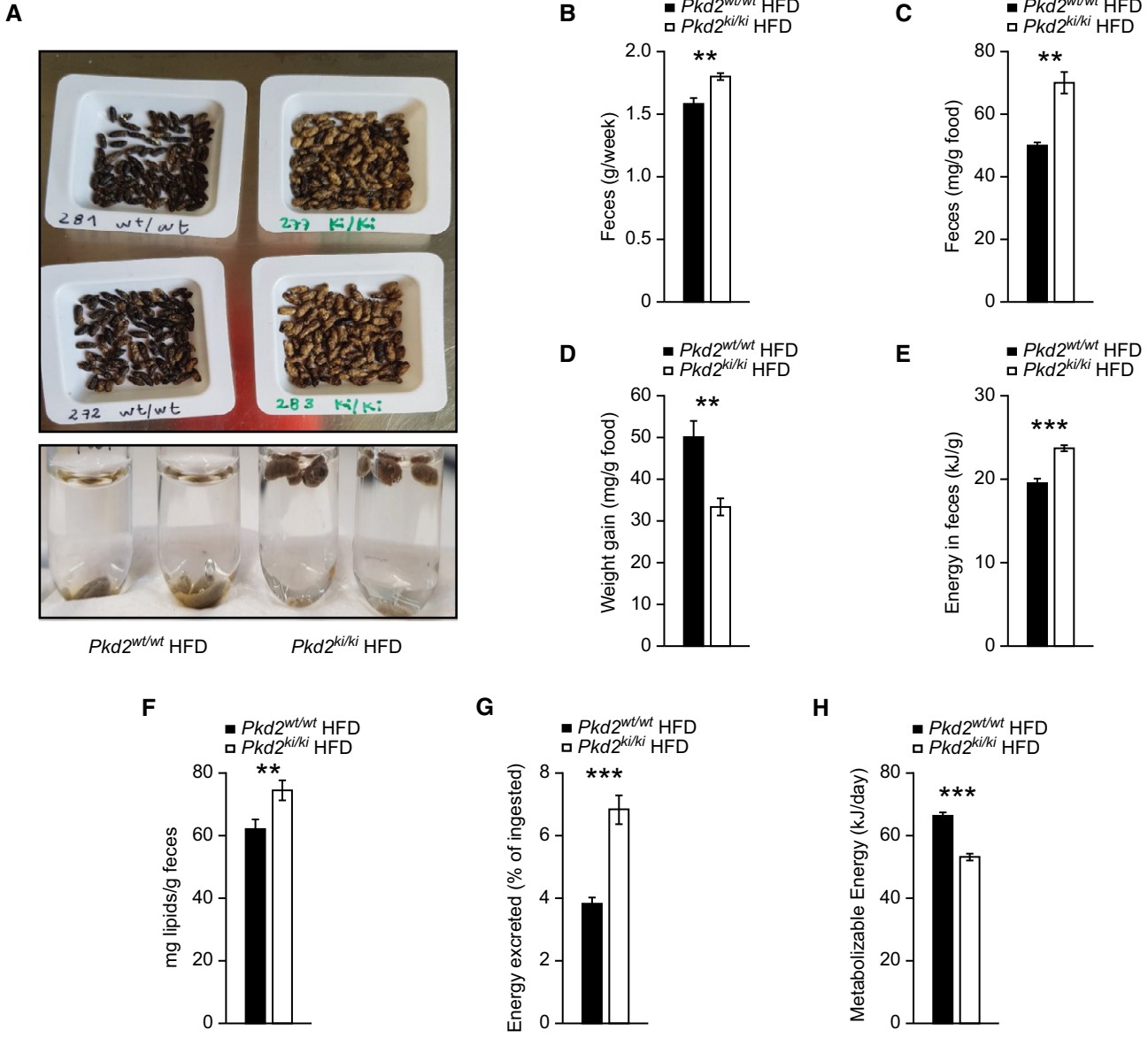

**Figure 2. *Pkd2* knockin mice excrete more energy in feces and present lower absorption of lipids in the intestine.**

A Pictures of feces collected from male mice fed HFD and photo of them placed in water.
B Weight of feces collected in a week.
C Feces excreted per gram of food consumed.
D Body weight gained per gram of food consumed.
E, F Energy content (E) and lipid content (F) per gram of feces.
G Percentage of energy excreted in feces from the total energy ingested.
H Metabolizable energy calculated from intake minus excreted and assuming a urinary excretion of 2%.

Data information: For panels (A–H), male mice after weaning were kept in individual cages during 2 weeks fed with HFD. For first 2 weeks, animals were acclimatized in the cages, and then during two more weeks, mice were monitored for food consumption and feces were analyzed for deposition of lipids and energy. $n = 7$ for $Pkd2^{wt/wt}$ and $n = 8$ for $Pkd2^{ki/ki}$. Data presented as average $\pm$ SEM. **$P \leq 0.01$, ***$P \leq 0.001$. Significances were assessed by using a two-tailed Student's $t$-test for independent groups.

nutrient assimilation by the organism (Woting & Blaut, 2018). However, as revealed by the dextran uptake assay, the inactivation of PKD2 did not affect intestinal permeability (Fig EV3D). Also, the total length of the gut was not affected in $Pkd2^{ki/ki}$ mice (Fig EV3E). Altogether, these observations suggested that PKD2 promotes

intestinal TG uptake by acting directly in enterocytes. To test this hypothesis, we utilized a Transwell culture system containing confluent Caco2 cell monolayers, a polarized human enterocyte-like cell line. Silencing of *Pkd2* in polarized Caco2 cells by two independent shRNA sequences (Fig EV3F and G) markedly reduced the release of

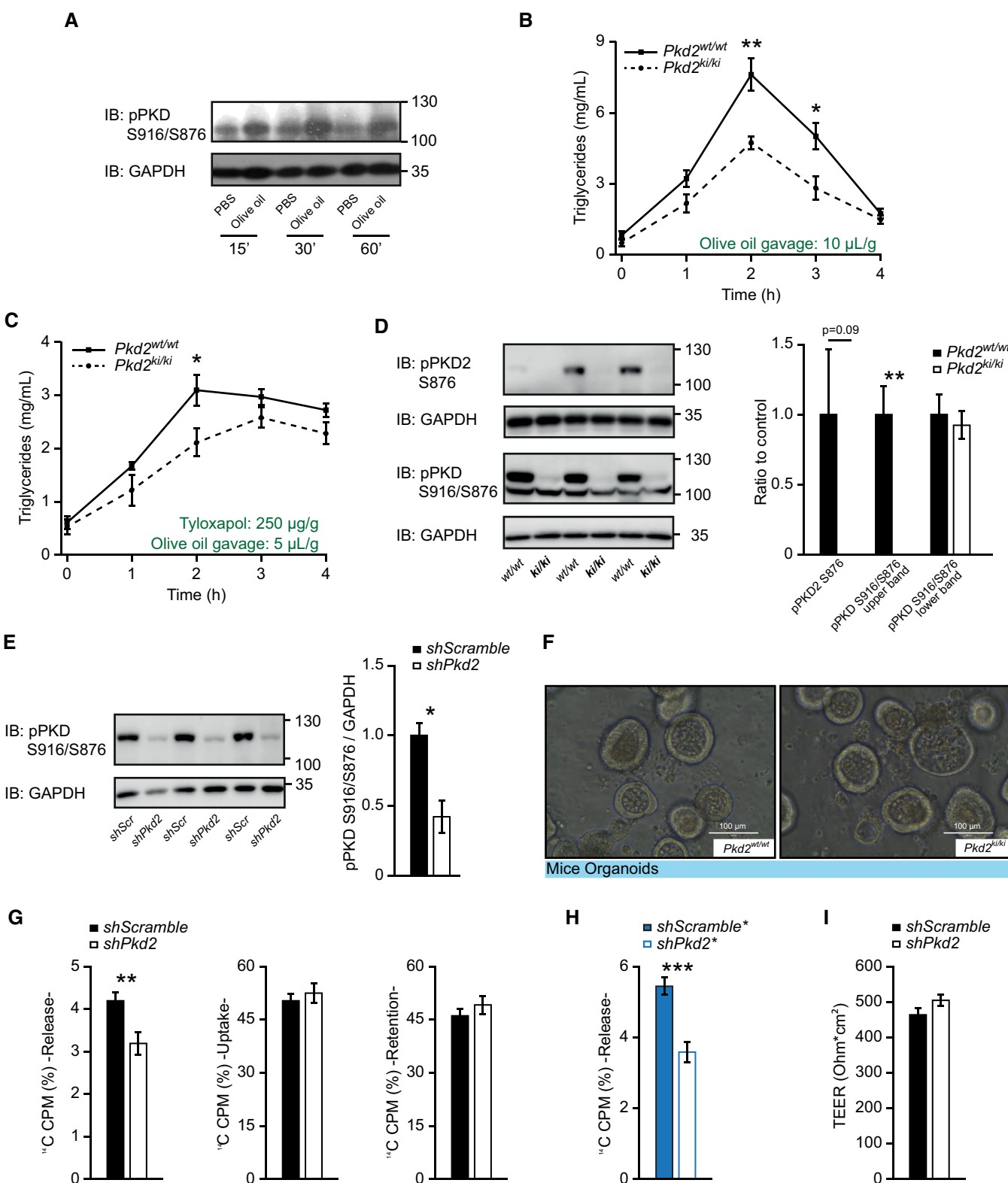

**Figure 3.**

**Figure 3. PKD2 inactivation decreases lipids' absorption.**

A   Western blot analysis of phosphorylated PKD (S916/S876) in small intestine of male C57BL/6 mice after overnight fasting, olive oil gavage (or PBS in controls), and dissection at different time-points.
B   Lipid tolerance test after oral gavage of olive oil (10 μl per gram of body weight). $n = 6$.
C   Lipid tolerance test 30 min after tyloxapol i.p. injection (250 μg/g body weight). In this case, only 5 μl of olive oil per gram of body weight was gavaged orally. $n = 6$.
D   Western blot analysis and quantification of phosphorylated PKD2 (S876) and phosphorylated PKD (S916/S876) in small intestine of $Pkd2^{wt/wt}$ and $Pkd2^{ki/ki}$ male mice. $n = 3$.
E   Western blot analysis and quantification of phosphorylated PKD (S916/S876) in Caco2 cells expressing *shscramble* or *shPkd2*. $n = 3$.
F   Representative light microscopy pictures of organoids derived from jejunum crypts of indicated male mice.
G   Percentage of released, uptaken and retained $^{14}$C-labeled fatty acids from Caco2 cells *shscramble* or *shPkd2* (sequence 1) in a Transwell system. $n = 6$.
H   Percentage of released $^{14}$C-labeled fatty acids by Caco2 cells expressing *shscramble*\* or *shPkd2*\* (sequence 2) grown in a Transwell system. $n = 6$.
I   Transepithelial electric resistance of the Caco2 cells *shscramble* or *shPkd2* (sequence 1). $n = 6$.

Data information: For panels (B and C), after acclimation, male mice after were treated as specified and triglycerides in circulation measured at the specified time-points. Data presented as average ± SEM, \*$P \leq 0.05$, \*\*$P \leq 0.01$, \*\*\*$P \leq 0.001$. Significances were assessed by using a two-tailed Student's $t$-test for independent groups.
Source data are available online for this figure.

TG into the basolateral side of the cells' monolayer (Fig 3G and H). However, the absence of PKD2 in cultured enterocytes did not influence FFA uptake from the apical side, retention of lipids inside the cells or paracellular permeability (Figs 3G and I and EV3H). Since PKD3 is also expressed in the intestine (Fig EV3I), we transfected Caco2 cells with shRNA against *Pkd3* (Fig EV5G). However, silencing of *Pkd3* did not affect TG uptake, retention, or release (Fig EV3J and K). These data indicate that the decrease in TG absorption is due to a release defect, which is caused specifically by the inactivation of PKD2. Our results so far indicate that the inactivation of PKD2 reduces the capacity of enterocytes to re-synthesize and/or to secrete TG, without affecting intestine morphology, cellularity, and development.

**Intestinal deletion of PKD2 reduces lipid absorption and protects from HFD-induced obesity**

Next, we tested whether PKD2 promotes TG absorption by acting autonomously in the intestine. To this end, we generated mice deficient for PKD2 in the gut by breeding $Pkd2^{flox/flox}$ mice (Ishikawa *et al*, 2016b) with Villin promoter-driven Cre animals ($Pkd2^{gutΔ/Δ}$ mice) (Madison *et al*, 2002). As determined by qPCR (Fig EV4A) and Western blot (Fig EV4B), $Pkd2^{gutΔ/Δ}$ mice lacked PKD2 protein in the small intestine but not in other organs tested. Of note, the deletion of PKD2 in intestine resulted in the lack of activation specific to PKD2 (upper band; Fig 4A) and marked reduction in total PKD

activity (Fig 4B). In agreement with the results obtained in the $Pkd2^{ki/ki}$ mice, olive oil ingestion in fasted $Pkd2^{gutΔ/Δ}$ mice (which were previously fed HFD for 6 weeks) evoked markedly lower circulating TG levels relative to the corresponding control $Pkd2^{flox/flox}$ mice (Fig 4C). Moreover, in line with the previous results from mice missing PKD2 activity globally, deletion of PKD2 specifically in the intestine resulted in resistance to HFD-induced obesity and body fat accumulation (Fig 4D and E) and improved glucose tolerance (Fig 4F). Moreover, circulating FFA and TG levels were reduced in mice deficient for PKD2 specifically in the intestine (Fig 4G and H). In addition, gut PKD2 deletion did not affect food intake or energy expenditure of mice (Fig EV4C and D) as well as intestine length or morphology, and hepatic architecture (Fig EV4E–G). Taken together, deletion of PKD2 specifically in the intestine is sufficient to limit lipid absorption from consumed food and protect against diet-induced obesity and glucose intolerance.

The composition of microbiota in the small intestinal is highly dependent on the type of diet consumed and interaction with the cells of the host. On the other hand, microorganisms colonizing the intestine are shown to be of functional relevance in lipid metabolism (Turnbaugh *et al*, 2006). To assess genotype-specific changes in small intestinal and cecal microbial composition in $Pkd2^{gutΔ/Δ}$ mice and respective $Pkd2^{flox/flox}$ control animals that were fed HFD for 2 weeks, we analyzed small intestinal scrapings and cecal content using 16S rRNA gene sequencing. Both genotypes were co-housed to obviate cage effects. Even though coprophagy potentially

**Figure 4. Intestinal deletion of PKD2 reduces lipid absorption, protects from HFD-induced obesity, and improves microbiota profile.**

A      Western blot and quantification of phosphorylated PKD (S916/S876) in small intestine of $Pkd2^{flox/flox}$ and $Pkd2^{gutΔ/Δ}$ male mice. $n = 3$.
B      Western blot and quantification of phosphorylated PKD (S744–748) in small intestine of $Pkd2^{flox/flox}$ and $Pkd2^{gutΔ/Δ}$ male mice. $n = 3$.
C      Lipids tolerance test after oral gavage of olive oil to $Pkd2^{flox/flox}$ and $Pkd2^{gutΔ/Δ}$ male mice after 6 weeks in HFD. $n = 10$.
D      Body weight gain of male mice of indicated genotypes in HFD. $n = 10$.
E      Quantification of fat, free fluid, and lean mass by nuclear magnetic resonance of mice in panel (B). $n = 10$.
F      Glucose tolerance test of male mice with indicated genotypes after 16 weeks in HFD. $n = 10$.
G, H  (G) Free fatty acids and (H) Triglyceride content in serum of male mice after 17 weeks in HFD. $n = 10$.
I      Differences in bacterial alpha diversity as indicated by Shannon Index. $n = 7$.
J      Changes in specific operational taxonomic unit (OTU) in small intestinal samples. $n = 7$.

Data information: Data presented as average ± SEM. In panels (I and J), central band indicates the median, the lower and upper parts of the box indicate the 25 and 75 percentiles and the whiskers connect data points outside these quartiles. \*$P \leq 0.05$, \*\*$P \leq 0.01$, \*\*\*$P \leq 0.001$. Significances in panels (A–H) were assessed by using a two-tailed Student's $t$-test for independent groups. In panel (G), statistical testing was performed using Mann–Whitney $U$-test. In panel (H), statistical testing was performed using PERMANOVA with corrections for multiple testing using the Benjamini and Hochberg method.
Source data are available online for this figure.

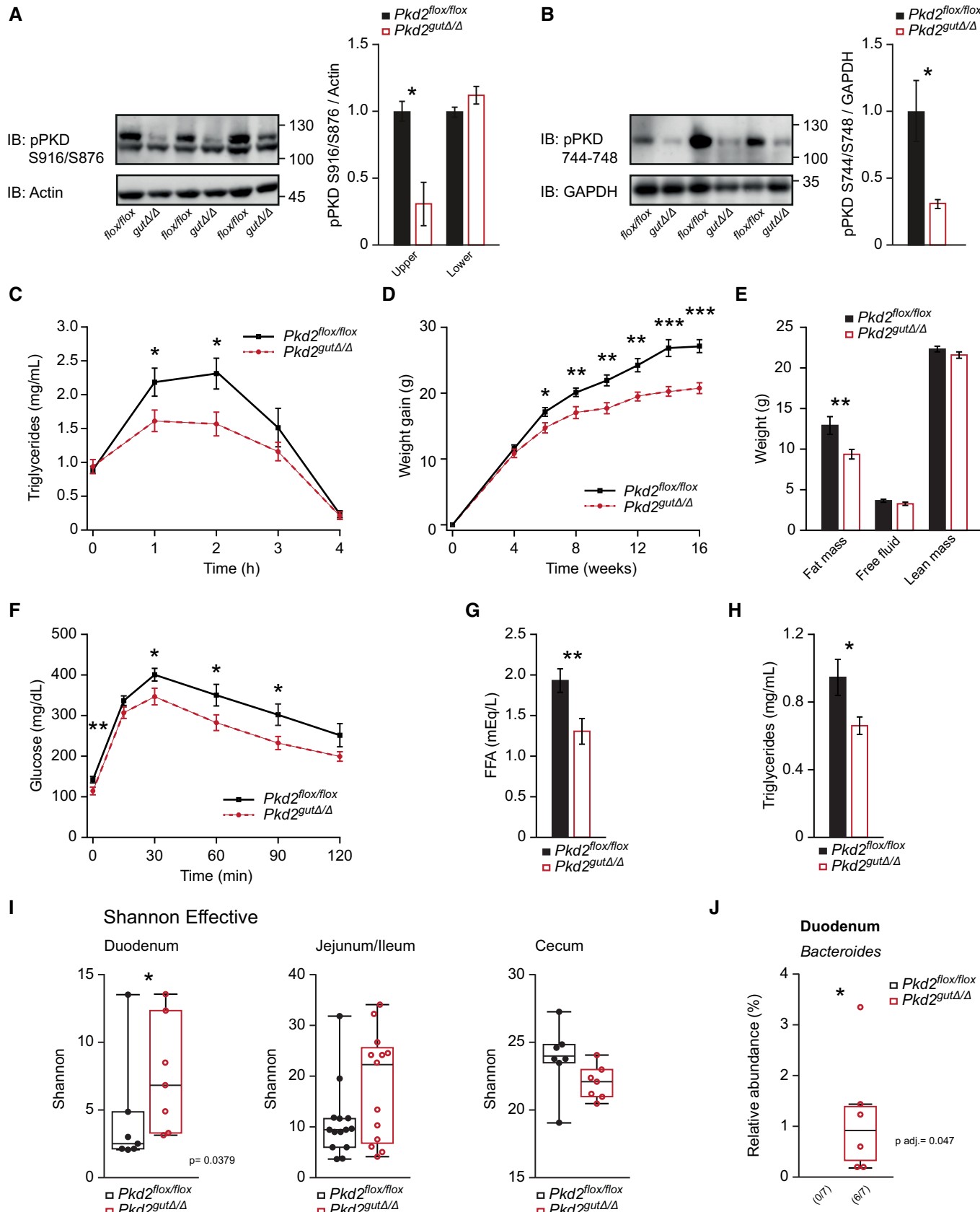

**Figure 4.**

weakens genotype-specific differences, we could identify significant changes in the microbial compositions. As indicated by Shannon Index, the alpha diversity of the microbiota species was significantly higher in $Pkd2^{gut\Delta/\Delta}$ mice than in control animals in the duodenum. A similar trend was observed in ileum and jejunum, but not in the cecum (Fig 4I). Supervised clustering yielded similar bacterial compositions in Jejunum and Ileum, which we, therefore, merged for further analysis (Fig EV4H and I). Analysis of beta diversity showed significant differences in duodenum and jejunum/ileum between the genotypes with no significant alterations in the cecum (Fig EV4J). Next, we analyzed specific changes in bacterial operational taxonomic units (OTUs) in the different gut segments. In duodenum, we identified members of bacteroides, which are associated with weight loss (Turnbaugh *et al*, 2006), to be only present in $Pkd2^{gut\Delta/\Delta}$ mice, but completely absent in control animals (Fig 4J). Taken together, we revealed the deletion of PKD2 in the intestine improves microbiota profile in mice fed HFD.

## PKD2 determines chylomicron size

Upon FFA and monoglyceride uptake by enterocytes, a series of enzymatic reactions leads to the TG re-synthesis, packing of TG into newly formed pre-chylomicrons at the ER, and their assembly with lipoproteins including APOB48, APOA1, and APOA4 (Mansbach & Siddiqi, 2016). To identify the mechanism by which PKD2 promotes TG output from enterocytes, we analyzed the expression and abundance of major factors involved in chylomicron formation. MOGAT2 and DGAT1 are responsible for TG re-synthesis; however, their expression or the protein levels of MOGAT2 were not affected by PKD2 inactivation (Figs 5A and EV4K). Similarly, the levels of MTTP, responsible for TG packing into pre-chylomicrons, were not affected by PKD2 (Figs 5A and EV4K). Likewise, the expression and the protein abundance of the lipoproteins associated with chylomicrons, APOB48 and APOA1, were not affected by PKD2 inactivation in the intestine (Figs 5A and EV4K). Interestingly, the abundance of APOA4 protein was markedly elevated in intestine deficient for PKD2 activity or PKD2 deletion (Fig 5B and C), while the expression of *Apoa4* was not affected by PKD2 inactivation (Fig EV4K). The abundance of the FA's transporter CD36 was also unchanged in the intestine isolated from $Pkd2^{ki/ki}$ mice (Fig 5A). APOA4 levels were also elevated in serum from $Pkd2^{ki/ki}$ mice, while APOB48 and

APOA1 levels were not affected by PKD2 inactivation (Fig 5D). Similarly, APOA4 levels were higher in the basolateral fraction secreted by PKD2-depleted Caco2 cells in the Transwell system (Fig 5E). APOA4 has been proposed to regulate chylomicron size and to mediate TG secretion by enterocytes (Kohan *et al*, 2013). Therefore, we next determined the size of chylomicrons in the blood of mice deficient for PKD2 activity. Indeed, the size of circulating chylomicrons was markedly reduced in $Pkd2^{ki/ki}$ mice relative to control animals (Fig 5F and G). Since PKD2 inactivation resulted in elevated APOA4 protein but did not affect its expression, PKD2 might regulate APOA4 levels and function by a posttranslational mechanism. Importantly, an *in vitro* kinase assay revealed that PKD2 phosphorylates APOA4 and that the PKD-specific inhibitor CRT0066101 abrogates PKD2-dependent phosphorylation (Fig 5H). Interestingly, in the presence of APOA4, the phosphorylation of PKD2 was also increased in the *in vitro* kinase assay, suggesting that interaction of PKD2 with this substrate might increase its auto-phosphorylation. Taken together, these results suggest that PKD2 promotes TG packing into chylomicrons presumably by directly phosphorylating and reversing the reported inhibitory role of APOA4 in chylomicrons biogenesis. While future studies will have to address the precise role of PKD2-mediated phosphorylation on APOA4 abundance, high levels of APOA4 observed in the absence of PKD2 are predicted to reduce the size of chylomicrons and thereby lipid transfer.

## Inhibition of PKD2 by a small-molecule compound ameliorates diet-induced obesity and diabetes

Our data suggest that pharmacological inhibition of PKD2 might restrict lipid uptake in the intestine and mitigate diet-induced obesity. In fact, the specific PKD inhibitor CRT0066101 has been generated and tested for *in vitro* and *in vivo* purposes (Harikumar *et al*, 2010) and it is a promising therapeutic agent for the treatment of several human pathologies (Harikumar *et al*, 2010; Thrower *et al*, 2011; Borges *et al*, 2015; Venardos *et al*, 2015; Yuan *et al*, 2017; Li *et al*, 2018; Sua *et al*, 2019). We found that treatment of Caco2 cells with CRT0066101 decreased TG output in a dose-dependent manner (Fig 6A). Interestingly, CRT0066101 did not decrease TG output in Caco2 cells depleted from PKD2 (Fig 6B). Next, we have treated mice with an oral dose of 10 mg/kg of CRT0066101 inhibitor daily.

**Figure 5. PKD2 promotes chylomicron size.**

A   Western blot (WB) analysis of specified proteins in small intestine of $Pkd2^{wt/wt}$ and $Pkd2^{ki/ki}$ male mice after 1 week of HFD. Quantification of the bands for each protein normalized to loading control and relative to $Pkd2^{wt/wt}$. $n = 3$.

B   WB of apoliprotein A4 in jejunum of $Pkd2^{wt/wt}$ and $Pkd2^{ki/ki}$ male mice after 1 week of HFD. Quantification of bands normalized to loading control and relative to $Pkd2^{wt/wt}$. $n = 3$.

C   WB of apoliprotein A4 in jejunum of $Pkd2^{flox/flox}$ and $Pkd2^{gut\Delta/\Delta}$ male mice after 1 week of HFD. Quantification of bands normalized to loading control and relative to $Pkd2^{flox/flox}$ $n = 7$.

D   WB analysis of specified apolipoproteins in 10 μl of serum of overnight fasted, 2 h refed male animals after 1 week in HFD. Quantification of bands relative to $Pkd2^{wt/wt}$. $n = 4$.

E   WB of 20 μl of basal medium of Caco2 cells expressing *shscramble* or *shPkd2* in Transwell system after 12 h of stimulation with oleic acid. $n = 6$.

F   Electron microscopy pictures of chylomicrons obtained by ultracentrifugation of plasma from $Pkd2^{wt/wt}$ and $Pkd2^{ki/ki}$ male mice as specified in methods.

G   Quantification of the diameter of the chylomicrons shown in panel (F). $n = 3$.

H   WB with specified antibodies after an *in vitro* kinase assay.

Data information: Data presented as average ± SEM, *$P \leq 0.05$, **$P \leq 0.01$, ***$P \leq 0.001$. Significances were assessed by using a two-tailed Student's *t*-test for independent groups.

Source data are available online for this figure.

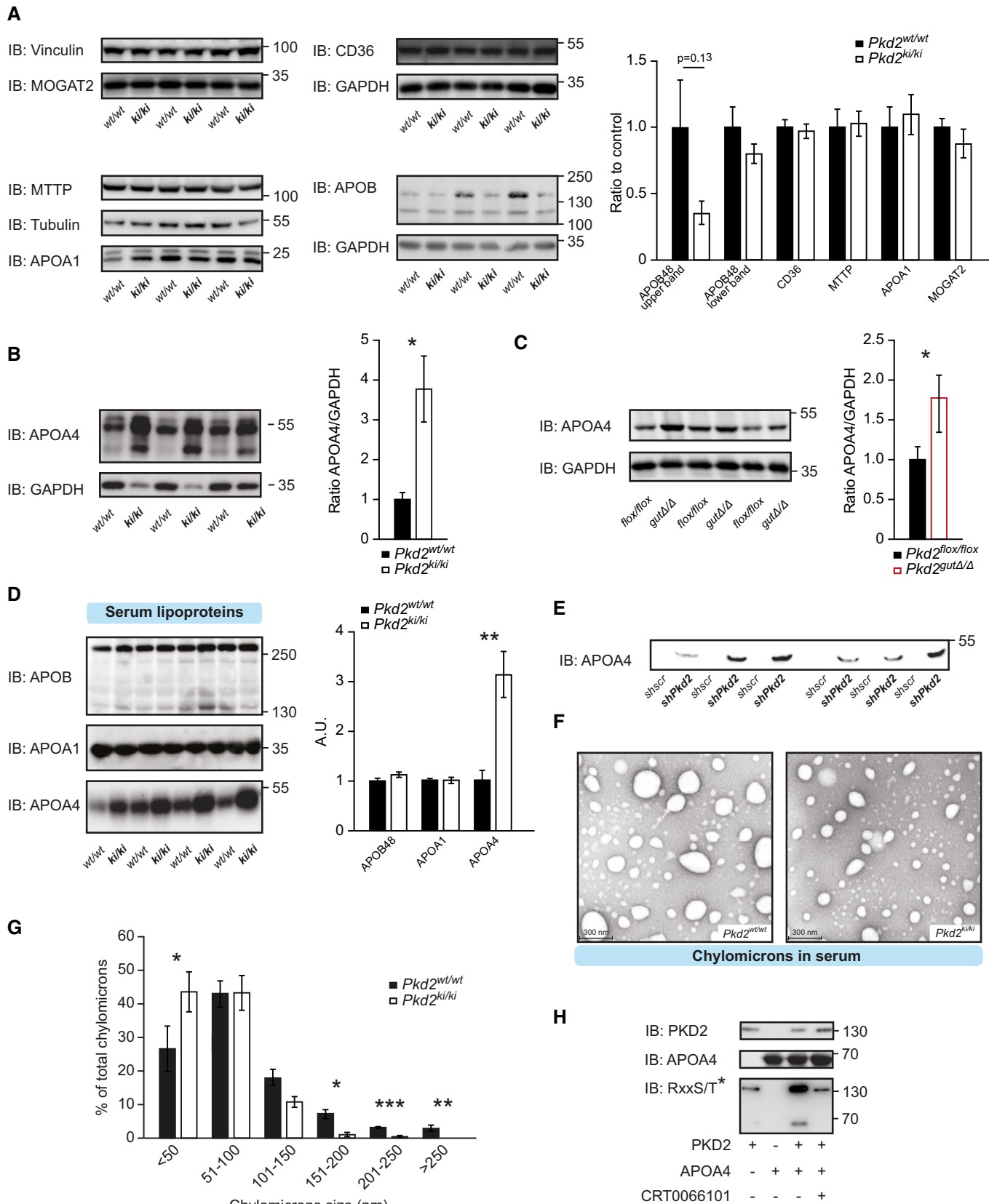

Figure 5.

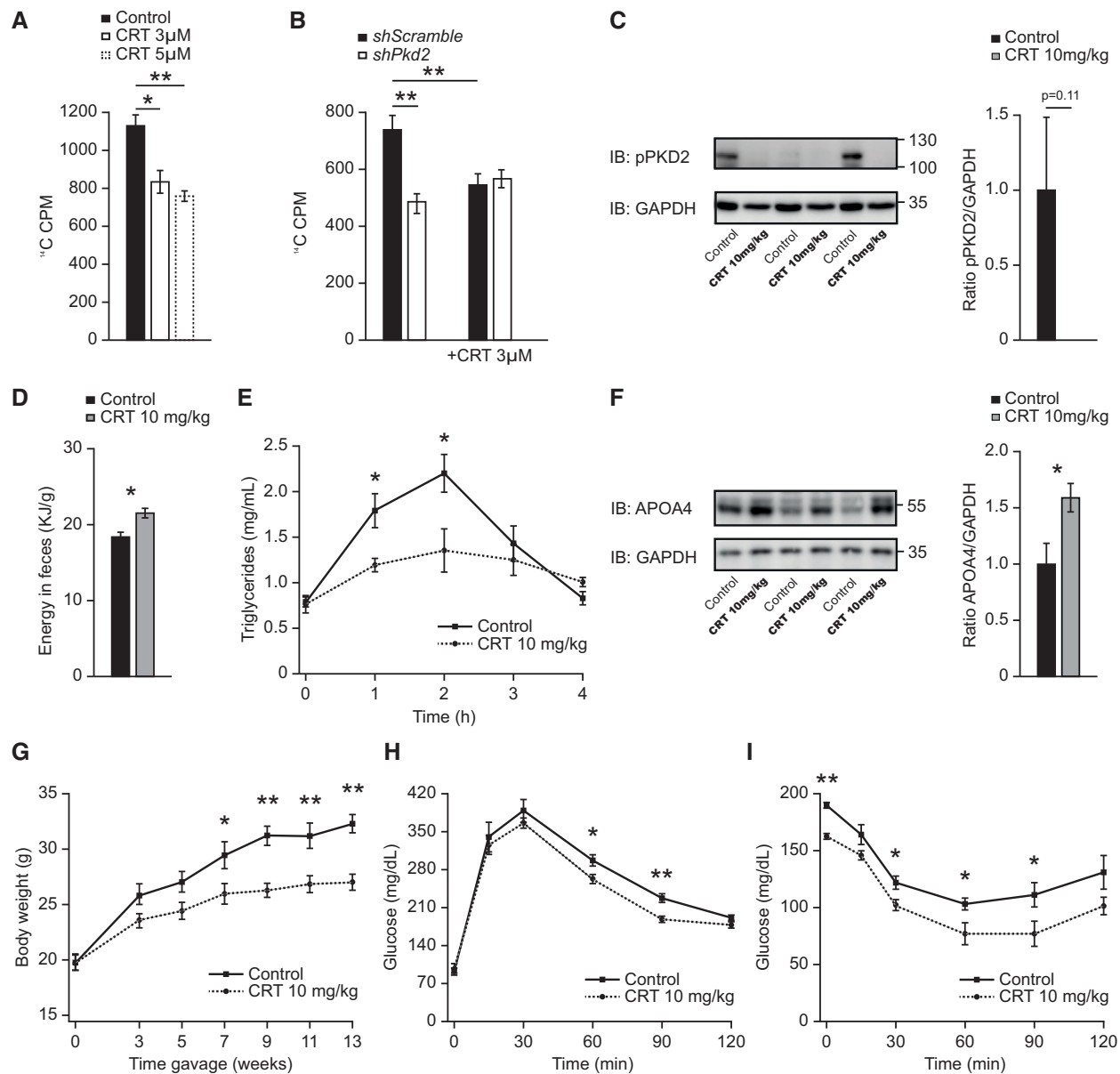

**Figure 6. Inhibition of PKDs by small-molecule compound reduces the degree of diet-induced obesity and diabetes.**

A  Basolateral release of $^{14}$C-labeled fatty acids in Caco2 cells grown in a Transwell system and treated with CRT0066101 at the indicated concentrations for 24 h. $n = 6$.

B  Basolateral release of $^{14}$C-labeled fatty acids in Caco2 cells expressing *shscramble* or *shPkd2* in a Transwell system and treated with CRT0066101 for 24 h. $n = 8$.

C  Western blot analysis of phosphorylated PKD2 in jejunum of male C57BL/6 mice in HFD and treated with CRT0066101 or control (water) for 13 weeks. $n = 3$.

D  Energy content per gram of feces of male mice in HFD and receiving daily gavage of CRT0066101 or water as control. $n = 6$ for control and $n = 5$ for CRT0066101.

E  Lipid tolerance test after oral gavage of olive oil to male mice after 9 weeks in HFD and receiving daily CRT0066101 or control. $n = 6$ for control and $n = 5$ for CRT0066101.

F  WB of Apolipoprotein A4 (APOA4) in jejunum of male C57BL/6 mice in HFD and treated with CRT0066101 or control (water) for 13 weeks. Quantification of bands normalized to GAPDH and relative to control (water) $n = 6$.

G  Body weight of male C57BL/6 mice in HFD and treated with CRT0066101 or control (water) for 13 weeks. $n = 6$ for control and $n = 5$ for CRT0066101.

H  Glucose tolerance test in male C57BL/6 mice which received CRT0066101 or water (control). Measurements after 10 weeks of treatment and HFD feeding. $n = 6$ for control and $n = 5$ for CRT0066101.

I  Insulin tolerance test in male C57BL/6 mice which received CRT0066101 or water (control). Measurements after 11 weeks of treatment and HFD. $n = 6$ for control and $n = 5$ for CRT0066101.

Data information: Data presented as average $\pm$ SEM, *$P \leq 0.05$, **$P \leq 0.01$. Significances in panels (A and B) were assessed by one-way ANOVA followed by the *post hoc* Tukey test. In panels (C–I), significances were assessed by using a two-tailed Student's *t*-test for independent groups.

Source data are available online for this figure.

At this dose of CRT0066101, PKD activity was diminished in the intestine but not in the liver or adipose tissue (Figs 6C and, EV5A and B). Importantly, treatment of mice fed HFD with CRT0066101 resulted in a higher deposition of energy in feces (Fig 6D), decreased TG absorption in the intestine (Fig 6E), and increased the intestinal abundance of APOA4 (Fig 6F). Interestingly, long-term treatment of mice fed HFD with PKD inhibitor resulted in a significantly lower body weight gain relative to the corresponding vehicle-treated (water) control animals (Fig 6G). However, oral administration of CRT0066101 inhibitor did not affect food intake, energy expenditure, or voluntary movements (Fig EV5C–E) as expected from the data obtained in the PKD2-deficient animals. Accordingly, treatment with CRT0066101 also improved HFD-evoked glucose and insulin intolerance and decreased insulin levels (Figs 6H and I, and EV5F). Administration of PKD inhibitor did not affect markers of liver function or hepatic TG accumulation but favored a healthier multilocular brown adipose tissue (Fig EV5G–I).

Next, we tested whether inhibition of PKD2 could be used to ameliorate previously established obesity. For this purpose, we first fed mice for 7 weeks with a HFD and then administered either CRT0066101 or a control solution (water) upon continued HFD feeding. PKD inhibitor substantially reduced body weight gain, which was associated with lower adiposity as well as improved glucose tolerance and insulin sensitivity (Fig 7A–E). Of note, the permeability of the intestine was not affected by the administration of PKD inhibitor (Fig EV5J). Taken together, these findings show that inhibition of PKD decreases TG absorption in the intestine and therefore protects from the development or exacerbation of obesity and associated diabetes.

As indicated by our data, inhibition of PKDs (especially PKD2) in the intestine might be an attractive strategy to ameliorate obesity in humans. Of note, the activity of PKD2 in the intestine of obese patients correlates positively with the levels of triglycerides and the percentage of glycated hemoglobin (HbA1c) in circulation (Figs 7F–H and EV5K). Moreover, there seems to be an inverse correlation between intestinal PKD2 activation and HDL levels in these patients, however, it did not reach statistical significance (Fig 7I). These indicate that the observations we have made in rodents, regarding the PKD2 function in lipid absorption from the intestine, might be valid in humans.

## Discussion

Our experiments suggest that PKD2 promotes lipid uptake in the intestine by acting directly in enterocytes. Accordingly, ablation of PKD2 function was associated with resistance to obesity and diabetes as well as improved microbiota profile in the intestine. Our experiments in human Caco2 cells revealed that FFA uptake and re-esterification, as well as retention of TG in the enterocyte, are not acutely affected by PKD2. However, PKD2 promotes the packing of TGs into chylomicrons. In fact, the absence of PKD2-dependent signaling in mice enterocytes resulted in smaller chylomicrons. At the molecular level, PKD2 regulated phosphorylation of APOA4, one of the major lipoproteins associated with the biogenesis of chylomicrons. We hypothesize that this event promotes the lipidation of chylomicrons and prevents their premature release. In fact, we have shown that PKD2 inactivation leads to elevated levels of APOA4 in

serum. APOA4 has been previously associated in humans and rodents with reduced extend of atherosclerosis and diabetes (Qu *et al*, 2019). The impact of APOA4 on chylomicron-mediated TG secretion has long been a subject of debate. APOA4 is primarily synthesized in the intestine, and its expression is highly induced in response to fat ingestion (Kohan *et al*, 2013). Genetic ablation of *Apoa4* in mice results in larger chylomicrons but does not overtly affect TG absorption in the intestine (Kohan *et al*, 2012; Kohan *et al*, 2013). On the other hand, overexpression of APOA4 in newborn pig enterocytes promotes chylomicron size and TG release (Lu *et al*, 2002; Lu *et al*, 2006). Moreover, overexpression of APOA4 in the intestine of mice resulted in higher postprandial TG levels in mice fed standard as well as HFD (Aalto-Setälä *et al*, 1994). These data indicate that an abundance of APOA4 does not simply correlate with its function. In the absence of PKD2-dependent signaling, we observed elevated APOA4 protein levels, both in the intestine and in the serum. However, the expression of *Apoa4* was not altered by PKD2 in any system tested. Therefore, APOA4 abundance is presumably regulated by PKD2-dependent phosphorylation of this protein. Another possible function of PKD2-dependent phosphorylation of APOA4 might be the regulation of APOA4 and chylomicron retention at ER. This model is suggested by studies utilizing APOA4 mutant carrying an ER retention tag. The expression of this mutant in a hepatic cell line or COS cells resulted in reduced secretion of APOB containing lipid particles (Gallagher *et al*, 2004; Weinberg *et al*, 2012). Similarly, unphosphorylated APOA4 might be trapped at the ER level. However, further studies will be needed to identify the precise mechanism of APOA4 regulation by PKD2. Alternatively, intestinal PKD2 may increase lipid absorption, chylomicron formation, and release as well as body weight through regulation of the stability/activity of lipid droplet-destined proteins in the Golgi (Kim *et al*, 2020). We should also add that, while inhibition/deletion of PKD2 did not affect FA uptake at the apical side of Caco2 cell layer in the short-term, it did lead to fattier stools in mice in the long term. This suggests a complex and dynamic role for PKD2 in regulation of the uptake and/or release of FA from enterocytes, which will be a subject of future studies.

Overconsumption of energy-dense diets is the major cause for the development of obesity. Generally, lipids are the most energetic components of the diet; therefore, increased fat consumption often promotes a positive energy balance. The capacity of the digestive system to absorb lipids increases in response to the elevated fat content in the diet (Petit *et al*, 2007; Hernández Vallejo *et al*, 2009; Clara *et al*, 2017). This process involves stimulation of expression and function of digestive enzymes (which degrade TG into FFAs, monoglycerides, and glycerol) and components of the enzymatic machinery responsible for FFA uptake, TG re-synthesis, packing of TG into chylomicrons, and their secretion into the lymphatic circulation (Petit *et al*, 2007; Hernández Vallejo *et al*, 2009; Clara *et al*, 2017). While a reduction in fat absorption in the intestine represents an attractive strategy to mitigate weight gain, limited pharmacological strategies exist targeting enzymatic machinery implicated in fat processing in the intestine. Orlistat and related drugs, targeting the activity of pancreatic lipase, are currently approved for the treatment of obese patients who are refractory to lifestyle interventions (Pilitsi *et al*, 2019). However, these drugs have limited efficacy during prolonged treatment and their usage is often discontinued because of associated side effects such as oily stools, oily spotting,

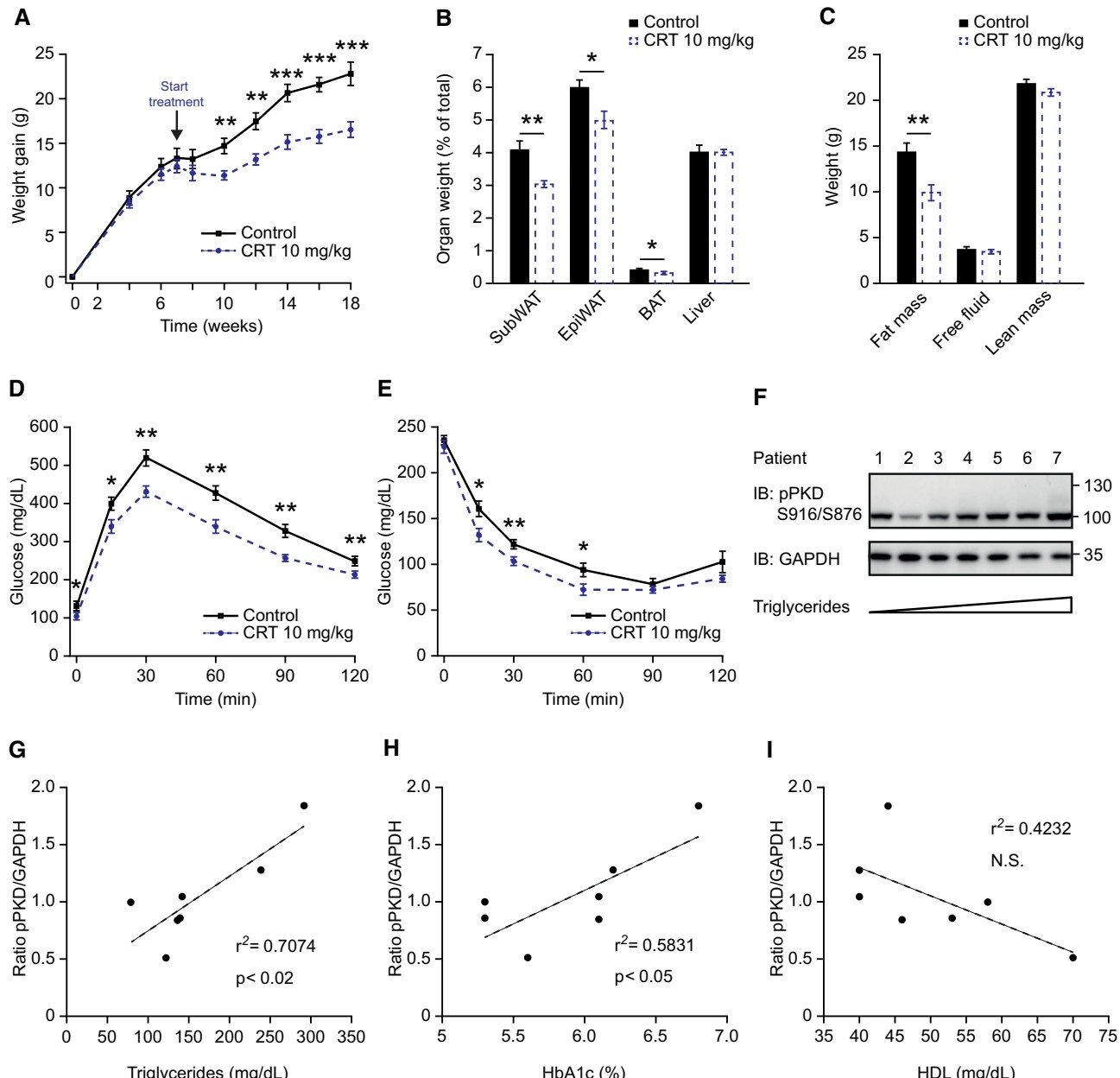

**Figure 7. Reduction in PKD activity in intestine ameliorates pre-established obesity in mice and correlates with healthier blood profile in obese patients.**

A   Body weight gain of male C57BL/6 mice that, after 7 weeks in HFD, started receiving treatment with CRT0066101 or water. HFD continued along with treatment. $n = 8$.

B   Organ weight (expressed as percentage of total body weight) of different fat depots and liver of male mice in panel (A).

C   Quantification of fat, free fluid, and lean mass by nuclear magnetic resonance (NMR) of male mice in panel (A).

D   Glucose tolerance test 8 weeks after starting the treatment of mice in panel (A).

E   Insulin tolerance test 10 weeks after starting the treatment of mice in panel (A).

F   Western blot analysis of PKD phosphorylation in proximal jejunum of morbidly obese female patients. Samples are loaded according to fasting triglyceride levels (lower to higher). $n = 7$.

G   Scatter plot of PKD phosphorylation in jejunum (normalized to GAPDH) vs. triglycerides of morbidly obese female patients. $n = 7$.

H   Scatter plot of PKD phosphorylation in jejunum (normalized to GAPDH) vs. percentage of glycated hemoglobin (HbA1c) of morbidly obese female patients. $n = 7$.

I   Scatter plot of PKD phosphorylation in jejunum (normalized to GAPDH) vs. high-density lipoprotein (HDL) of morbidly obese female patients. $n = 7$.

Data information: Data presented as average $\pm$ SEM, $*P \leq 0.05$, $**P \leq 0.01$, $***P \leq 0.001$. Significances were assessed by using a two-tailed Student's *t*-test for independent groups. In panels (G–I), $r^2$ = square of the Pearson product–moment correlation coefficient. $p$ determined for Pearson correlation coefficient for the given degrees of freedom.

Source data are available online for this figure.

fecal urgency, fecal incontinence, hyper-defecation, and flatus with discharge (Pilitsi *et al*, 2019). Moreover, orlistat and related drugs do not protect from the cardiovascular complications often associated with obesity and they ameliorate T2D only to a limited extent. Inhibition of PKDs in the intestine of mice or deletion and inactivation of PKD2 in animals led to similar protection from body weight gain as described previously for orlistat (Murtaza *et al*, 2017; Zhao *et al*, 2018), importantly, while we found that PKD2 inactivation or inhibition of PKDs increased stool lipid content, without any associated overt gastrointestinal disturbances or symptoms. A direct comparison of both drugs under standardized conditions would provide an overview of their relative efficiency and side effects.

Our data indicate that deletion of PKD2 was associated with positive changes in gut microbiota. Intestinal dysbiosis is defined as a loss of microbial diversity and specific phyla and is linked to obesity, diabetes, and cardiovascular disease (Henao-Mejia *et al*, 2012; Qin *et al*, 2012; Wilkins *et al*, 2019). Our data suggest improved stability of microbial diversity in the absence of PKD2. Furthermore, we detected changes in specific bacterial taxa assigned to the genus bacteroides in the duodenum. Bacteroides have been reported to decrease in different models of obesity and in response to high-fat diet, indicating that deletion of PKD2 prevents nutritional perturbations of the microbiota (Turnbaugh *et al*, 2006; Martinez-Guryn *et al*, 2018). However, at this point, we cannot determine whether the observed changes in the microbiota composition are at least partially the cause or the consequence of the observed reduction in TG absorption. Therefore, even though the mechanisms linking PKD2 to the changes in microbiota composition remain elusive, the intestinal deletion of PKD2 seems to positively affect the microbiota diversity.

Previous studies indicate that inhibition of PKD might also lead to beneficial effects on other pathologies often associated with obesity. The inactivation of PKD using specific inhibitors improves heart function (Venardos *et al*, 2015), reduces the incidence of pancreatitis (Thrower *et al*, 2011; Yuan *et al*, 2017), and prevents progression of different cancer types (Harikumar *et al*, 2010; Borges *et al*, 2015; Li *et al*, 2018; Sua *et al*, 2019). Inhibition of the PKD pathway might also lead to increased energy dissipation by adipose tissue (Löffler *et al*, 2018). However, it was previously reported that oral administration of CRT0066101, in a dosage 8 times higher than the one we used in our experiments, reached peak concentrations in pancreatic tumors after 2 h and decreased PKD1/2 activation by 50% (Harikumar *et al*, 2010); this raises the possibility that effective inhibition of PKD, at our relatively low dosage, was only achieved in the GI-tract. In fact, the oral administration of an inhibitor of all the PKD family members did not result in inhibition of PKDs in any other organs than the intestine. This might explain the fact that we observed only a marginal increase in energy expenditure in mice treated with PKD inhibitor. Alternatively, decreased food efficiency, which results in a reduction of net energy supply, might decrease energy dissipation and normalize it to the level observed in control animals.

On the other hand, several previous reports have indicated that inhibition of PKDs might lead to some adverse effects. For instance, increased accumulation of lipids and fibrosis in the liver altered pancreatic β cell function or increased permeability of the intestine (Sumara *et al*, 2009; Xiong *et al*, 2016; Xiao *et al*, 2018; Mayer *et al*, 2019; Zhang *et al*, 2020). Nevertheless, inhibition of PKDs in mice

resulted in better glucose tolerance and insulin sensitivity, likely secondary to decreased body weight. Also, we did not observe increased intestinal permeability or altered hepatic enzymes/histology in mice with inactivation of PKD2 or in animals treated with a PKD inhibitor. Although PKD inhibition did not evoke any obvious adverse effects in mice, the development of drugs targeting PKD2 only might increase the safety of the potential anti-obesity therapy.

Altogether, we postulate that inhibition of PKD-dependent signaling in the intestine might serve as a potential targeted therapeutic strategy for obesity and associated diseases. As an exploratory study, we have analyzed intestinal samples of obese patients and found convincing evidence of how PKD activation correlates with TG in circulation and other metabolic parameters. These results strengthen our previous findings; however, potential side effects must still be carefully assessed.

Taken together, we have identified PKD2 as a central kinase mediating fat absorption in the intestine and propose that PKD2-dependent phosphorylation of APOA4 might determine chylomicron lipidation and biogenesis. Importantly, we verified that the inhibition of PKD2 might be a valid strategy for the treatment of obesity and associated diseases in the future. Finally, we have confirmed that PKD2 plays a similar role in the regulation of enterocytes function in mice, human cell line, and the intestine of humans.

## Materials and Methods

### Generation of mouse models

All animal experiments were approved by the local institutional animal care (Regierung von Unterfranken, Germany) and conducted according to the guidelines and state regulations. Experiments were performed under animal protocol numbers AK 55.2-2531.01-124/13 and 55.2-2532-2-741. Mice were maintained in a specific pathogen-free facility with the ambient temperature set at 23°C, following a 12-h light–dark cycle and given ad libitum access to water and standard chow diet which was exchanged under indicated experimental conditions to high-fat diet (HFD). All mice were closely monitored by the authors, facility technicians, and an independent veterinary scientist responsible for animal welfare. Euthanasia was performed by cervical dislocation, in a separate area, away from other animals, and all efforts were made to minimize suffering. No animals died or became ill during the development of this research.

*Pkd2* knockin mice were obtained from The Jackson Laboratories (Matthews, Navarro *et al*, 2010b). These mice were maintained in a C57BL/6 background and presented point mutations in Ser[707] and Ser[711] which were mutated to alanines (*Pkd2$^{ki/ki}$*). The generation of mice bearing a specific deletion of intestinal PKD2 was achieved by cross-breeding Villin-Cre mice (B6.Cg-Tg(Vil1-cre)1,000 Gum/J) with *Pkd2* flox/flox mice (Ishikawa *et al*, 2016a). Genotyping was performed for indicated mice models following standard PCR protocols and using respected primer sets from Appendix Table S1.

### Animal experiments

Mouse body weight development was closely monitored and reported here on a weekly basis when mice are under the specified diet. All experiments were performed in male mice.

## Oral administration of PKD inhibitor

Protein kinase D inhibitor CRT0066101 was dissolved in water (10 mg/kg) and administered orally to the mice by daily gavage, while the control mice were gavaged with water. Mice were maintained under the specified diets and had ab libitum access to food and water. C57BL/6 mice were administered CRT inhibitor treatment at 4 weeks of age, while for the rescue experiment, inhibitor was given under the same circumstances but treatment started only after 7 weeks of HFD feeding. Intestinal barrier permeability was further assessed 1 h after oral gavage of FITC-Dextran 4.

## Indirect calorimetry

Energy expenditure, food intake, and activity measurements were obtained in a PhenoMaster (TSE Systems) as previously described (Trujillo Viera *et al*, 2016; El-Merahbi *et al*, 2020). Briefly, mice were kept in separate metabolic cages with unlimited access to water and to the specified diet. After 48 h of acclimation, data were collected every 10 min including photobeam breaks, oxygen consumption, carbon dioxide dissipation, and food and water consumption. Results in bars represent the average of each night/day cycle.

## Metabolic tests

A glucose tolerance test (GTT) was performed after overnight fasting. Mice blood glucose was assessed before and 15, 30, 60, 90, and 120 min after intraperitoneal injection of glucose (2 g/kg). For the insulin tolerance test (ITT), the same measurements were performed after 4 h of fasting and intraperitoneal injection of 0.5 U/ kg of recombinant insulin. For glucose measurements, one blood drop was drawn from the tail tip into a test strip and measured with a glucometer (Accu-Chek® Roche).

Glucose-stimulated insulin secretion was measured with the Ultra Sensitive Mouse Insulin ELISA Kit (Crystal Chem) according to the manufacturer's instructions. Briefly, overnight fasted mice were i.p. injected with 30% glucose (3 g/kg), and blood samples were taken from the tail tip at 0, 2, 5, 15, and 30 min. 5 μl of serum was added to the antibody-coated microplate and incubated. Then, the second reaction with the anti-insulin enzyme conjugate and the third reaction with the enzyme substrate were performed before measuring absorbance at 450 and 630 nm. Calculations were performed according to the Low Range Assay (0.1–6.4 ng/ml) standards.

Lipids tolerance test was performed as described previously (Wang *et al*, 2016b). Briefly, overnight fasted mice were gavaged with olive oil (10 μl/g body weight). Blood was collected at indicated times for measurement of serum triglycerides content. For lipid absorption under tyloxapol treatment, 250 μg/g of body weight was i.p. injected to mice 30 min before the gavage of olive oil 5 μl/g body weight.

Triglycerides, free FAs, and glycerol in circulation were determined in serum samples using the mentioned kits and according to their manufacturers' instructions.

Liver triglycerides were measured by homogenization of 50 mg of liver in 500 μl of lysis buffer and lipid extraction in methanol:chloroform, phase extraction, and evaporation. The lipids were redissolved in DMSO and triglycerides quantified with the mentioned kit and according to the manufacturers' instructions.

## Feces analysis

The bedding of single-caged mice was collected, and the ingested food ingestion from the individual mice was assessed. All the stools were manually collected. After overnight dissecation at 60°C, the samples were weighted and frozen for further analysis. The amount of feces produced per mouse was compared with the amount of food consumed and the weight gain. About 3 g of dried feces material was homogenized, and 1 g was pressed into a tablet and accurately weighted. Calorie content from feces and diet was measured in duplicates in a 6400 Automatic Isoperibol Calorimeter (Parr Instrument Company). To analyze energy excreted, the amount of energy in feces was compared with the amount of energy ingested (the product between grams of food and caloric content of the food). Lipid extraction from feces was performed as previously described (Kraus *et al*, 2015).

## DNA extraction and 16S rRNA Sequencing

We collected cecal contents and mucosal scrapings from the small intestine, immediately snap froze in liquid nitrogen, and stored them at −80°C. DNA isolation, library preparation, and sequencing were performed by the ZIEL—Core Facility Microbiome of the Technical University of Munich. Briefly, DNA was extracted using previously published protocols (Klindworth *et al*, 2013). For the assessment of bacterial communities, primers specifically targeting the V3-V4 region of the bacterial 16S rRNA (primers in Appendix Table S1) gene were used including a forward and reverse illumine-specific overhang and a barcode. Sequencing was performed using an Illumina MISeq DNA platform. Multiplexed sequencing files were analyzed using the IMNGS platform, based on the UPARSE approach for sequence quality check, chimera filtering, and cluster formation (Edgar, 2013). For the analyses, standard values for barcode mismatches, trimming, expected errors, and abundance cutoff were used and we only included sequences between 300 and 600 bp for analyses. Downstream analyses of the IMNGS platform output files were performed using the RHEA R pipeline (Lagkouvardos *et al*, 2017). In brief, we normalized the abundances and assessed the quality of sequences using rarefaction curves (McMurdie & Holmes, 2014). We performed the analysis of alpha and diversity beta diversity as well as group comparison using default settings with an exception for the group comparisons, where we excluded alpha diversity from the analysis and set the prevalence cutoff value to 0.5. For presentation, we modified the obtained graphical output using inkscape (https://inkscape.org). Assignment of OTUs to taxons was performed using the SILVA database (Quast *et al*, 2013).

## Cell culture and stable cell lines

Caco2 cells were cultured in Dulbecco's modified eagle's medium 4.5 g/L glucose (DMEM), 10% fetal bovine serum (FBS), non-essential amino acids (NEAA), 1 mM sodium pyruvate, and 40 μg/ml gentamycin. For the generation of shRNA cell lines, *Pkd2* shRNA lentiviral sequences or scramble (non-targeting) was first cloned

into the pLKO.1-puro vector then packed into viral particles in packaging cells (HEK293T) using 3rd-generation packaging vectors. Caco2 cells were spinfected with viral supernatant then selected with puromycin, and the deletion efficiency was further assessed by Western blot.

## Lipid uptake *in vitro*

Lipid transport studies were assessed in Caco2 cells seeded into Transwells. $1.5 \times 10^5$ cells were seeded into a 12-well cell culture insert of 1-μm pore size Transwell and maintained in for 14 consecutive days with media change every 2 days. For the lipid uptake and release, the apical side of the membrane was loaded with radioactive $^{14}$C-palmitic acid (0.2 μCi), oleic acid (1 mM), and taurocholic acid (2 mM) in DMEM high glucose 10% FBS. The basal medium contained DMEM high glucose 0.1% FBS. After 24 h of incubation, cell lysates and the apical and basolateral medium were collected and the $^{14}$C-palmitic acid was measured in a scintillation counter for aliquots of apical and basal medium and for cell lysate. The same protocol was used in Caco2 cells that were treated with CRT0066101 (3 and 5 μM) for 24 h (added together with the mentioned radioactive FA cocktail).

## Transepithelial electrical resistance

Barrier integrity of the established Caco-2 *in vitro* models was determined by chopstick hand electrode (Millicell ERS-2, Millipore, Billerica, MA) as previously described by Srinivasan *et al* (2015). TEER (transepithelial electrical resistance) was indicated as $\Omega*cm^2$ for model setup with *shScramble* and *shPkd2* Caco-2 cells after 14 days of differentiation to enterocyte-like Caco-2 cells. The permeability of Caco-2 cells monolayer was further assessed by using fluorescein isothiocyanate dextran 4,400 (FD-4) as the model compound for paracellular tight junction transport.

## *In vitro* kinase assay

Recombinant human proteins used for the study, PKD2 and APOA4, were both purchased from Abcam. Kinase reactions were carried out in a kinase reaction buffer containing the immune complex, recombinant proteins, and ATP as described in Bedford *et al* (2011). Then, Western blot was performed and incubated with a primary antibody against the phosphorylated motif (RxxS/T*; Cell Signaling).

## Western Blot

Proteins from tissues and cell culture were extracted with RIPA buffer supplemented with phosphatase and protease inhibitor (PPI, Thermo Fisher Scientific). Protein concentration was measured by Pierce™ BCA Protein Assay Kit (Thermo Fisher Scientific) following the manufacturer's protocol. Reduced protein extracts were separated on 10% SDS–PAGE by electrophoresis and transferred to PVDF membranes under wet transfer. Membranes were blocked in 5% (w/v) milk in TBS-T before overnight probing with indicated primary antibodies at 4°C, followed by TBS-T washes and incubation with corresponding secondary antibody. The signals were detected with an enhanced chemiluminescence solution in an

Amersham Imager 680 (GE Healthcare) and automatically quantified by the software according to the manufacturer's instructions.

## Chylomicron analysis

A study of chylomicrons size was performed as previously reported (Wang *et al*, 2016b) with some modifications. Mice in HFD were overnight fasted and given a bolus of olive oil (10 μl/g of body weight) 2 h before sampling blood. Plasma from three mice was pooled, and 200 μl was diluted with saline 1:4 ratio. The mixture was centrifuged for 3 h at 117,000 *g* and 4°C. The top layer containing the chylomicrons was then removed, and 5 μl was applied to the carbon-coated copper grids. Chylomicrons were stained with uranyl acetate for 15 min and visualized under electron microscopes Zeiss EM 900 and 80 kV (50,000×). Quantification of the chylomicrons size was done with ImageJ from 4 different pictures per genotype.

## Mice organoid culture

For a generation of mice organoids, 1–2 cm of jejunum from $Pkd2^{wt/wt}$ and $Pkd2^{ki/ki}$ mice was washed with HBSS and cut open. Mucus and villi were removed with HBSS and by scraping with glass slides. Then, a series of six washings with cold HBSS and shaking (in order: vortex 5 s, rotation 30 min, inversion, and 3× manual shaking) ensure the removal of the villi and extraction of the crypts. After centrifugation, crypts are seeded in Matrigel™ (5,000 crypts/ml) in 50 μl drops. After an incubation period of 10–20 min at 37°C, 300 μl culture medium was added to cover the solidified drops. Basal medium, for resuspension of pellets, contains DMEM-F12 advanced complemented with 1× N2, 1× B27 w/o vitamin A, 1× anti–anti, 10 mM HEPES, 2 mM GlutaMAX-I, and 1 mM *N*-acetylcysteine. The organoids were maintained in basal medium supplemented with 500 ng/ml hR-Spondin 1, 100 ng/ml rec Noggin, 50 ng/ml hEGF, 3 μM CHIR99021, and 1 mM valproic acid (also 10 μM Y-27632 only the first day after splitting). Medium was changed every 2 days.

## Histological analysis

For organoid imaging studies, mice organoids were collected, and after removal of the Matrigel® matrix (Corning), they were fixed with 4% paraformaldehyde for 1 h at 4°C and resuspended in Histogel™ Drops were embedded in paraffin, sectioned, and deparaffinized. Antigen retrieval was performed with a steamer and blocking with 5% normal serum. Organoids are then incubated with primary antibodies against indicated antigens followed by fluorescent labeling with appropriate secondary antibodies. Samples were mounted using Fluoromount-G™ with DAPI. Slides were visualized using Leica TCS SP8 confocal microscope (Sato *et al*, 2009).

Adipose tissues, liver, and intestine were dissected and directly placed to paraformaldehyde 4% at 4°C for 24-h fixation. Tissues were dehydrated. Samples were embedded in paraffin and cut into 5-μm sections. Standard hematoxylin–eosin staining was performed, and digital pictures were taken in a Leica light microscope DM4000B at 20× for adipose tissue, 40× for liver sections, and 10× for intestine. A blinded experiment was performed to measure adipocyte size. Six specimens per genotype and about 400–500 adipocytes per specimen were measured with ImageJ for that purpose.

For pancreatic islet stainings and quantification, the pancreas was dissected and directly embedded into OCT. Three different sections (7 μm) per subject were taken with a distance of 50 μm. Sections were fixed (4% PFA) and blocked (5% BSA, 0.2% Triton in PBS) before overnight incubation (4°C) with the primary antibody. Then, the sections were washed and incubated with the secondary antibody (1 h at RT) before mounting with DAPI.

**Real-time PCR analysis**

Total RNA was extracted from tissues and cells using QIAzol Lysis Reagent (QIAGEN) according to the manufacturer's instructions. 1 μg of RNA was reversely transcribed with the First Strand cDNA Synthesis Kit (Thermo Fisher Scientific) according to their protocol. cDNA was then diluted 1:15 and used for qPCR with SYBR Green. The absolute quantification of PKD isoform copy numbers was performed according to a standard protocol from Applied Biosystems. In brief, primers were designed to be located within same exons and genomic DNA of known concentration was used for creating a standard curve reflecting copy numbers. Forward and reverse primers used according to Appendix Table S1.

**Human samples**

Fasting blood samples were collected from 7 morbidly obese (BMI 42.7–54 kg/m$^2$) female patients aged 33–57 years prior to Roux-en-Y gastric bypass (RYGB) surgery for standard biochemical analysis. A whole-wall segment of proximal jejunum (5 cm) was collected from fasted patients during the surgery and gently rinsed in ice-cold PBS before rapidly snap-freezing in liquid nitrogen. At the time of surgeries, two patients were on metformin medication for type 2 diabetes and one patient for polycystic ovary syndrome. Prior to surgeries, all patients observed a standard 2–4 week very low-calorie diet (1,000 kcal/day) which comprised 2 daily liquid meals (such as egg white shakes or vegetable soup) and 1 daily solid meal (such as tuna fish or chicken breast salads) that were low in fat and carbohydrates and rich in protein (100g). Informed, written consent was obtained from all patients, and usage of human material was approved by the Ethics Committee of the Medical Faculty of the University of Würzburg (approval number EK AZ 188/17—MK). All the human experiments were performed according to the principles of the WMA Declaration of Helsinki and the Department of Health and Human Services Belmont report.

**Statistical analyses**

The results are presented as mean values ± standard error of the mean (SEM). Significances were assessed by using a two-tailed Student's *t*-test for independent groups. *P* values of 0.05 or lower were considered as statistically significant (*$P < 0.05$, **$P < 0.01$, and ***$P < 0.001$).

For microbiota studies, analysis of alpha diversity was performed with GraphPad prism v6 using the non-parametric Mann–Whitney *U*-test. We visualized beta diversity using non-metric multidimensional scaling based on generalized UniFrac and tested for significance using PERMANOVA. Differences in OTUs were determined using Fisher's exact test with adjustments for multiple testing using the Benjamini and Hochberg method.

**The Paper Explained**

**Problem**
Consumption of highly caloric diets rich in triglycerides is one of the major causes for the development of obesity and associated disorders. However, up to date, limited pharmacological strategies exist to inhibit lipid absorption in the intestine. We have investigated the function of the diacylglycerol-activated protein kinase D2 in the regulation of triglyceride absorption in the intestine and its impact on the development of obesity.

**Results**
We have shown that PKD2 is activated upon triglyceride ingestion in the intestine. Moreover, we have demonstrated that PKD2 phosphorylates APOA4 to promote chylomicron size, and therefore, triglycerides transport in the intestine. Importantly, deletion, inactivation, or inhibition of PKD2 in mice suppresses triglyceride absorption in the intestine and ameliorates obesity as well as associated diabetes. Additionally, deletion of PKD2 is associated with improved microbiota in the intestine. Finally, our results indicate that PKD2 activity correlates with triglyceride levels in patients, and silencing of PKD2 in human enterocytes reduces chylomicron-mediated triglyceride transport.

**Impact**
Our findings indicate that PKD2 is an attractive target for pharmacological intervention to limit lipid absorption in the intestine and therefore ameliorate obesity and prevent the development of diabetes.

For scatter plots of human samples, $r^2$ = square of the Pearson product–moment correlation coefficient and *p* was determined for Pearson correlation coefficient for the given degrees of freedom.

# Data availability

Datasets produced in this study are available at: PRJNA701875; https://www.ncbi.nlm.nih.gov/sra/PRJNA701875

**Expanded View** for this article is available online.

## Acknowledgements
We thank Dr. Annette Schuermann for help with experimental procedures and for the critical reading of the manuscript. We thank Prof. Dr. C. Stigloher, Claudia Gehrig-Höhn, and the Imaging Core Facility of Biocenter of the University of Würzburg for help with the electron microscopy experiments. We are grateful to Caroline Ziegler, PD Dr. Klaus Neuhaus, and Dr. Ilias Lagkouvardos from the ZIEL Core Facility Microbiome/NGS at the Technical University of Munich for the outstanding technical assistance with sample processing for high-throughput 16S rRNA gene amplicon sequencing and for the help with statistical analysis. Dr. Olga Sumara and Dr. Izabela Sumara for comments on the manuscript. This study was funded by European Research Council (ERC) Starting Grant SicMetabol (no. 678119), Emmy Noether Grant Su 820/1-1 from the German Research Foundation (DFG), EMBO Installation Grant from European Molecular Biology Organization (EMBO), the Dioscuri Centre of Scientific Excellence—The program initiated by the Max Planck Society (MPG), managed jointly with the National Science Centre, and mutually funded by the Ministry of Science and Higher Education (MNiSW) and the German Federal Ministry of Education and Research (BMBF), and Collaborative Research Centre 1371 (CRC) Microbiome Signatures—Functional Relevance in the Digestive Tract funded

by the German Research Foundation (DFG). We would like to thank Servier Medical Art by Servier (smart.servier.com) for sharing free images under a Creative Commons Attribution 3.0 Unported License. Some elements from sma rt.servier.com were used for creating the figures. Open Access funding enabled and organized by ProjektDEAL.

## Author contributions

GS and JT-V wrote the manuscript and were responsible for the experimental design. GS obtained the funding for the project. JT-V and RE-M performed most of the experiments. MM, EI, SY, MK, and MH contributed to the design of the experiments and helped with the interpretation of the data. VS, TK, AL-V, AS, SR, MN, MW, IH, SM, CF, AEM, MCL, and IW performed some of the experiments and helped with data analyses and acquisition. MR, AG, FS, and MH were involved in the collection of human samples.

## Conflict of interest

The authors declare that they have no conflict of interest.

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
