## [Review Process File · EMBO Molecular Medicine]

Protein Kinase D2 drives chylomicron-mediated lipid transport in the intestine and promotes obesity

Jonathan Trujillo-Viera, Rabih El-Merahbi, Vanessa Schmidt, Till Karwen, Angel Loza-Valdes, Akim Strohmeyer, Saskia Reuter, Minhee Noh, Magdalena Wit, Izabela Hawro, Sabine Mocek, Christina Fey, Alexander E. Mayer, Mona C. Löffler, Ilka Wilhelmi, Marco Metzger, Eri Ishikawa, Sho Yamasaki, Monika Rau, Andreas Geier, Mohammed Hankir, Florian Seyfried, Martin Klingenspor and Grzegorz Sumara **DOI: 10.15252/emmm.202013548**

Corresponding author: Grzegorz Sumara (grzegorz.sumara@uni-wuerzburg.de)

Review Timeline:

Submission Date:	6th Oct 20
Editorial Decision:	10th Nov 20
Revision Received:	12th Feb 21
Editorial Decision:	9th Mar 21
Revision Received:	16th Mar 21
Accepted:	17th Mar 21

Editor: Lise Roth

Transaction Report:

10th Nov 2020

Dear Dr. Sumara,

Thank you for the submission of your manuscript to EMBO Molecular Medicine. We have now received feedback from the three reviewers who agreed to evaluate your manuscript. As you will see from the reports below, the referees acknowledge the interest of the study and are overall supporting publication of your work pending appropriate revisions.

Addressing the reviewers' concerns in full will be necessary for further considering the manuscript in our journal, and acceptance of the manuscript will entail a second round of review. EMBO Molecular Medicine encourages a single round of revision only and therefore, acceptance or rejection of the manuscript will depend on the completeness of your responses included in the next, final version of the manuscript. For this reason, and to save you from any frustrations in the end, I would strongly advise against returning an incomplete revision.

When submitting your revised manuscript, please carefully review the instructions that follow below. Failure to include requested items will delay the evaluation of your revision:

- 1) A .docx formatted version of the manuscript text (including legends for main figures, EV figures and tables). Please make sure that the changes are highlighted to be clearly visible.
- 2) Individual production quality figure files as .eps, .tif, .jpg (one file per figure).
- 3) A .docx formatted letter INCLUDING the reviewers' reports and your detailed point-by-point responses to their comments. As part of the EMBO Press transparent editorial process, the point-by-point response is part of the Review Process File (RPF), which will be published alongside your paper.
- 4) A complete author checklist, which you can download from our author guidelines (<https://www.embopress.org/page/journal/17574684/authorguide#submissionofrevisions>). Please insert information in the checklist that is also reflected in the manuscript. The completed author checklist will also be part of the RPF.
- 5) Before submitting your revision, primary datasets produced in this study need to be deposited in an appropriate public database (see <https://www.embopress.org/page/journal/17574684/authorguide#dataavailability>). Please remember to provide a reviewer password if the datasets are not yet public. The accession numbers and database should be listed in a formal "Data Availability" section (placed after Materials & Method). Please note that the Data Availability Section is restricted to new primary data that are part of this study.

6) We would also encourage you to include the source data for figure panels that show essential data. Numerical data should be provided as individual .xls or .csv files (including a tab describing the data). For blots or microscopy, uncropped images should be submitted (using a zip archive if multiple images need to be supplied for one panel). Additional information on source data and instruction on how to label the files are available at

7) Our journal encourages inclusion of *data citations in the reference list* to directly cite datasets that were re-used and obtained from public databases. Data citations in the article text are distinct from normal bibliographical citations and should directly link to the database records from which the data can be accessed. In the main text, data citations are formatted as follows: "Data ref: Smith et al, 2001" or "Data ref: NCBI Sequence Read Archive PRJNA342805, 2017". In the Reference list, data citations must be labeled with "[DATASET]". A data reference must provide the database name, accession number/identifiers and a resolvable link to the landing page from which the data can be accessed at the end of the reference. Further instructions are available at .

8) We replaced Supplementary Information with Expanded View (EV) Figures and Tables that are collapsible/expandable online. A maximum of 5 EV Figures can be typeset. EV Figures should be cited as 'Figure EV1, Figure EV2" etc... in the text and their respective legends should be included in the main text after the legends of regular figures.

- Additional Tables/Datasets should be labeled and referred to as Table EV1, Dataset EV1, etc. Legends have to be provided in a separate tab in case of .xls files. Alternatively, the legend can be supplied as a separate text file (README) and zipped together with the Table/Dataset file. See detailed instructions here:

9) The paper explained: EMBO Molecular Medicine articles are accompanied by a summary of the articles to emphasize the major findings in the paper and their medical implications for the non-specialist reader. Please provide a draft summary of your article highlighting

10) For more information: There is space at the end of each article to list relevant web links for further consultation by our readers. Could you identify some relevant ones and provide such information as well? Some examples are patient associations, relevant databases, OMIM/proteins/genes links, author's websites, etc...

11) Author contributions: the contribution of every author must be detailed in a separate section (before the acknowledgments).

12) A Conflict of Interest statement should be provided in the main text

13) Every published paper now includes a 'Synopsis' to further enhance discoverability. Synopses are displayed on the journal webpage and are freely accessible to all readers. They include a short stand first (maximum of 300 characters, including space) as well as 2-5 one-sentences bullet points that summarizes the paper. Please write the bullet points to summarize the key NEW findings. They should be designed to be complementary to the abstract - i.e. not repeat the same text. We encourage inclusion of key acronyms and quantitative information (maximum of 30 words / bullet point). Please use the passive voice. Please attach these in a separate file or send them by email, we will incorporate them accordingly.

Please also suggest a striking image or visual abstract to illustrate your article. If you do please provide a png file 550 px-wide x 400-px high.

14) As part of the EMBO Publications transparent editorial process initiative (see our Editorial at <http://embomolmed.embopress.org/content/2/9/329>), EMBO Molecular Medicine will publish online a Review Process File (RPF) to accompany accepted manuscripts.

In the event of acceptance, this file will be published in conjunction with your paper and will include the anonymous referee reports, your point-by-point response and all pertinent correspondence relating to the manuscript. Let us know whether you agree with the publication of the RPF and as here, if you want to remove or not any figures from it prior to publication.

I look forward to receiving your revised manuscript.

Yours sincerely,

Lise Roth

Lise Roth, PhD
Editor
EMBO Molecular Medicine

To submit your manuscript , please follow this link:

Link Not Available

Photos 400-800 DPI

*Additional important information regarding figures and illustrations can be found at <https://bit.ly/EMBOPressFigurePreparationGuideline>

***** Reviewer's comments *****

Referee #1 (Remarks for Author):

In this manuscript, the authors describe a role for Protein Kinase D2 as a driver for chylomicron-mediate lipid transport in the intestine. The conclusion is that PKD2 represents a key signaling node promoting dietary fat absorption and may serve as a target for treatment of obesity. Although these findings are interesting, there are several concerns that need to be addressed.

Major concerns:

- 1- Figure 1 lacks controls such as an IHC for PKD2 and demonstration that the PKD2^{ki/ki} is indeed not active.
- 2- Both, PKD1 and PKD3 also have effects on obesity and insulin resistance. Since PKD enzymes can form complexes and cross-activate each other it could be that PKD2^{ki/ki} fails to activate PKD1 or PKD3 and this leads to/or contributes to the here described effects. How was this excluded?
- 3- In Figure EV2H-I the authors describe that the inactivation of PKD2 in the intestine did not affect expression of PKD1 and PKD3, and they conclude that this "excludes the possibility of any indirect function of other PKD family members in the intestine as a compensatory mechanism". I don't think this conclusion can be drawn: First, the analysis of mRNA does not indicate if/and how much protein is present. This should be clarified with immunohistochemistry. Moreover, indirect effects would rather be expected via PKD complex formation and cross-activation.
- 4- In Figure 3 the authors suggest that PKD2 activity in the intestine promotes TG absorption, but this is not really shown. Can wildtype mice be stimulated to induce TG absorption and PKD2 activity in the intestine be demonstrated in these mice (via IHC or otherwise)?
- 5- In Figure 5 A and C at least 3 mice per group should be analyzed. Data in Figures 5A-C also should be quantified to show statistical significance.
- 6- For the knockdown experiments it seems that there was only one shRNA sequence used. This is

not sufficient and a second shRNA sequence should be used to confirm data; or a rescue experiment should be performed.

7- Figure 6 implicates that "inhibition of PKD2 by a small molecule (CRT0066101) ameliorates diet-induced obesity and diabetes". However, CRT0066101 is a pan PKD inhibitor and it is completely unclear which PKD isoform was responsible for the effects observed. If the authors feel this figure is needed to support the studies, then they should provide an in depth analyses of which PKD enzymes have been affected.

Other points:

1- In figure 5B, samples should be normalized, such that that GAPDH shows similar levels.

2- The figure legends are not detailed enough.

3- In Figure 5G, the antibody that was used for detecting phosphorylation in the kinase assay is not specific for the PKD motif. Cell Signaling has a more specific antibody for this motif. This figure also lacks an IB for PKD2 to show input. The IB for RxxS/T* should show a larger molecular weight range in order to also show autophosphorylation of PKD2.

4- The RxxS/T* antibody is not described in Table S2.

Referee #2 (Remarks for Author):

This is an interesting manuscript describing the contribution of intestinal Pkd2 enzyme to diet induced obesity and glucose tolerance. By using several different mouse models where Pkd2 gene is deficient, inactive or inhibited by a pharmacological compound, the authors elegantly demonstrate that lipid transport through the intestine is altered and thus inhibition of Pkd2 protects from obesity. Mechanistically, they show that the ApoA4 protein levels are increased upon Pkd2 inactivation, probably secondary to decreased phosphorylation. They also show that gut microbiota play a role in this scenario. Finally, they demonstrate the druggability of this enzyme and its effect to protect against diet-induced obesity. This is a nice demonstration, with state-of-the-art approaches including organoids.

That being said, the reviewer has few comments for the authors.

Upon inactivation of Pkd2 (Pkd2ki/ki), the authors show that insulin secretion is not significantly modified upon a glucose challenge (fig. EV1H). Interestingly, it seems that basal insulin levels (at T0) are increased in Pkd2ki/ki compared to control, despite improved insulin sensitivity. Since the knock-in strategy might affect Pkd2 activity where this enzyme is expressed, including the pancreatic islets, is there any histological effects of Pkd2 inactivation in the pancreatic islets?

How is expressed ApoA4 in the intestine of mice deficient for Pkd2 (Pkd2gut Δ/Δ)? And upon pharmacological inhibition of Pkd2?

The gut microbiota analysis in Pkd2gut Δ/Δ mice is interesting. However, it is not clear whether this microbiota remodeling is a cause or consequence of Pkd2 gene deficiency and/or improvement of glucose homeostasis. This should be discussed.

The figure 6 nicely shows that blocking Pkd2 activity with CRT0066101 improves glucose homeostasis, decrease TG absorption, and can ameliorate glucose parameter after obesity is established. Although this is an interesting approach, it is not clear whether this pharmacological strategy is directly targeting intestinal Pkd2. The authors should demonstrate that the inhibitor effects are mediated through the inhibition of intestinal Pkd2. In addition, is there any effects of this inhibitor on insulin secretion during glucose tolerance test?

Minor : there are some typos, errors in the references, and errors in figure labelling (fig 4C->D).

Referee #3 (Remarks for Author):

Previous studies have shown that Protein Kinase D1 (PKD1) promotes obesity by inhibiting energy dissipation in adipocytes and PKD3 increases hepatic insulin resistance. The current study by Jonathan T. et al seeks to investigate an unexpected role of PKD2 in lipid homeostasis. The major findings of this study include: 1) that PKD2 enhances chylomicron-mediated TG transfer in enterocytes; 2) PKD2 increases chylomicron size to enhance the TG secretion on the basolateral side of the mouse and human enterocytes; 3) PKD2 phosphorylated APOA4 that is one of the apolipoproteins associated with chylomicrons; 4) Deletion, inactivation of PKD2 improved HFD-induced obesity, diabetes and improves the gut microbiota profile in mice; 5) Pharmacological inhibition of PKD2 improved HFD-induced obesity, diabetes and changed gut microbiota profile in mice. Therefore, the authors demonstrated that PKD2 is a key signaling molecule affecting dietary fat absorption and may serve as a potential target for treatment of obesity. It is an interesting study, however, there are several issues to be addressed.

Specific Comments:

1. The gender information of the mice used in the experiments should be included in both methods and figure legends.
2. Previous studies have found that protein kinase D1 deletion in adipocytes enhances energy dissipation and protects against adiposity. Does CRT 0066101 affect PKD1 activity?
3. In Figure EV1, Pkd2 ki/ki mice have smaller adipocyte sizes of EpiWAT and SubWAT compared to control group, does deletion of Pkd2 affect the browning of white adipose tissue or affect the mitochondrial biogenesis, or increase UCP1 expression?
4. Previous studies found that inhibition of PKD might lead to increased accumulation of lipids and fibrosis in the liver. It would be important to determine the levels of TG in livers of Pkd2 ki/ki mice and pharmacological inactivation of Pkd2 in mice.
5. CRT treatment reduced the weight of BAT in Figure 6F, does CRT treatment affect the histological features of BAT?
6. It is shown that Pkd2 ki/ki mice exhibited similar levels of ALT and AST, does CRT treatment affect the levels of AST and ALT?
7. "we analyzed specific changes in Bacterial operational taxonomic units (OTUs) in the different gut segments. In Duodenum, we identified members of Bacteroides, which are associated with weight loss (Turnbaugh et al., 2006), to be only present in Pkd2 gut Δ/Δ mice, but completely absent in control animals". It might be important to move these data from the Supplemental Material into the main text.
8. Figure 5F, why is there no error bar for this graph? Also please include the labels for the y-axis.
9. In the Discussion, the authors discuss potential therapies of inhibition of PKD2 and inhibition of PKD-dependent signaling in the intestine as a potential targeted therapeutic strategy for obesity and T2D, do T2D or obese patients have increased levels or activities of PKD2?

Minor:

Typo: "ThermoFischer" in Page 37.

Response letter EMM-2020-13548

Referee #1 (Remarks for Author):

“In this manuscript, the authors describe a role for Protein Kinase D2 as a driver for chylomicron-mediate lipid transport in the intestine. The conclusion is that PKD2 represents a key signaling node promoting dietary fat absorption and may serve as a target for treatment of obesity. Although these findings are interesting, there are several concerns that need to be addressed.”

We thank reviewer #1 for the enthusiastic assessment of our manuscript.

Major concerns:

“1- Figure 1 lacks controls such as an IHC for PKD2 and demonstration that the PKD2ki/ki is indeed not active.”

We have performed western blot for active PKD2 in the small intestine of PKD2 ki/ki and control animals. As presented in Fig 3D, the activity of PKD2 is almost completely abolished in the intestine from PKD2 ki/ki mice.

“2- Both, PKD1 and PKD3 also have effects on obesity and insulin resistance. Since PKD enzymes can form complexes and cross-activate each other it could be that PKD2ki/ki fails to activate PKD1 or PKD3 and this leads to/or contributes to the here described effects. How was this excluded?”

To exclude this possibility we have measured activation of PKD1 and PKD2 in the intestine of PKD2 ki/ki mice. While activity (phosphorylation of PKD2 on S876) was markedly reduced in the PKD2 ki/ki mice compared to control animals, the activity of PKD1 was not altered (Fig 3D), as assessed by an antibody that can recognize both: PKD1 phosphorylated

on S916 (lower band on the WB) and PKD2 phosphorylated on S876 (upper band on the WB).

“3- In Figure EV2H-I the authors describe that the inactivation of PKD2 in the intestine did not affect expression of PKD1 and PKD3, and they conclude that this "excludes the possibility of any indirect function of other PKD family members in the intestine as a compensatory mechanism". I don't think this conclusion can be drawn: First, the analysis of mRNA does not indicate if/and how much protein is present. This should be clarified with immunohistochemistry. Moreover, indirect effects would rather be expected via PKD complex formation and cross-activation.”

We agree with reviewer #1 that at this point we have over-interpreted our data. Therefore, to support this conclusion we have measured the protein abundance of PKD1, PKD2, and PKD3 in the intestine of PKD2 *ki/ki* and control mice (we have measured all the variants of these kinases with western blots since the existing antibodies do not work for immunohistochemistry in our hands). Interestingly, the levels of none of these proteins were altered in the absence of PKD2 activity (Fig EV2H). As stated above, the activity of PKD1 (phosphorylation on S916) was also not altered by the inactivation of PKD2 (Fig 3D). Unfortunately, the antibody which would allow us to trace the activity of PKD3 is not existing, therefore we cannot directly exclude that PKD3 activation is not altered in the absence of PKD2 activity. However, when PKD3 is knocked-out in Caco2 cells, there was no effect on lipids transport in the transwell system (Fig EV3K).

“4- In Figure 3 the authors suggest that PKD2 activity in the intestine promotes TG absorption, but this is not really shown. Can wildtype mice be stimulated to induce TG absorption and PKD2 activity in the intestine be demonstrated in these mice (via IHC or otherwise)?”

We have shown upon challenge with olive oil, that the activity of PKD2 increases in the intestine of wildtype mice (Fig 3A).

“5- In Figure 5 A and C at least 3 mice per group should be analyzed. Data in Figures 5A-C also should be quantified to show statistical significance.”

Following the suggestion of reviewer #1, we have extended our analyzes on samples isolated from new mice. The quantifications are available next to each WB picture

“6- For the knockdown experiments it seems that there was only one shRNA sequence used. This is not sufficient and a second shRNA sequence should be used to confirm data; or a rescue experiment should be performed.”

We have utilized another sequence of shRNA specific for PKD2. The results were almost identical to those observed before (Fig 3H and EV3G-H).

“7- Figure 6 implicates that "inhibition of PKD2 by a small molecule (CRT0066101) ameliorates diet-induced obesity and diabetes". However, CRT0066101 is a pan PKD inhibitor and it is completely unclear which PKD isoform was responsible for the effects

observed. If the authors feel this figure is needed to support the studies, then they should provide an in depth analyses of which PKD enzymes have been affected.”

We agree with reviewer #1 that the CRT0066101 inhibitor targets all the PKDs. Therefore we have changed “inhibition of PKD2” to “inhibition of PKDs” throughout the entire manuscript. Nevertheless, we think that the results presented in figure 6 are an integral part of our manuscript as our new results indicate the following:

- The dose of the CRT0066101 inhibitor used in this study suppresses activation of PKDs in the intestine but not in liver or adipose tissue, where previous studies established the function of PKD3 and PKD1 (Fig 6C and EV5A-B)
- Silencing of PKD3 in Caco2 cells did not influence the TG transport (Fig EV3K).
- Inhibition of PKDs using CRT0066101 inhibitor in Caco2 cells depleted from PKD2 did not further decrease TG transport (Fig 6B).

Taking all of these into consideration we postulate that CRT0066101 inhibitor protects against obesity primary by suppressing the activity of PKD2 in the intestine.

Other points:

“1- In figure 5B, samples should be normalized, such that that GAPDH shows similar levels.”

We have quantified for normalization as suggested by the reviewer.

“2- The figure legends are not detailed enough.”

We have included on the figure legends the number of mice and the gender of animals used for the experiments. We have also included a more detailed description of the experiments.

“3- In Figure 5G, the antibody that was used for detecting phosphorylation in the kinase assay is not specific for the PKD motif. Cell Signaling has a more specific antibody for this motif. This figure also lacks an IB for PKD2 to show input. The IB for RxxS/T should show a larger molecular weight range in order to also show autophosphorylation of PKD2.*

We have used the antibody against RxxS/T* motive as the other antibody from cell signaling recognizes only LxRxxS/T* motif and PKDs are known to phosphorylate I/V/LxRxxS/T* motif. Following the suggestion of reviewer #1, we have shown on the figure also the upper part of the western blot and included loading for PKD2 on the figure.

4- The RxxS/T antibody is not described in Table S2.”*

We have included it.

Referee #2 (Remarks for Author):

“This is an interesting manuscript describing the contribution of intestinal Pkd2 enzyme to diet induced obesity and glucose tolerance. By using several different mouse models where Pkd2 gene is deficient, inactive or inhibited by a pharmacological compound, the authors

elegantly demonstrate that lipid transport through the intestine is altered and thus inhibition of Pkd2 protects from obesity. Mechanistically, they show that the ApoA4 protein levels are increased upon Pkd2 inactivation, probably secondary to decreased phosphorylation. They also show that gut microbiota play a role in this scenario. Finally, they demonstrate the druggability of this enzyme and its effect to protect against diet-induced obesity. This is a nice demonstration, with state-of-the-art approaches including organoids.”

We thank reviewer #2 for the enthusiastic assessment of our manuscript.

“That being said, the reviewer has few comments for the authors. Upon inactivation of Pkd2 (Pkd2ki/ki), the authors show that insulin secretion is not significantly modified upon a glucose challenge (fig. EV1H). Interestingly, it seems that basal insulin levels (at T0) are increased in Pkd2ki/ki compared to control, despite improved insulin sensitivity. Since the knock-in strategy might affect Pkd2 activity where this enzyme is expressed, including the pancreatic islets, is there any histological effects of Pkd2 inactivation in the pancreatic islets?”

We have performed the histological analyzes and stained for insulin in pancreas isolated from PKD2 ki/ki and control mice. Even though, we observe non-significant changes in the mean fluorescence intensity of insulin, the area of the islet relative to the pancreas area is increased in *Pkd2^{ki/ki}* mice (Fig EV1F-H).

“How is expressed ApoA4 in the intestine of mice deficient for Pkd2 (Pkd2gutΔ/Δ)? And upon pharmacological inhibition of Pkd2?”

The APOA4 levels are also upregulated in the intestine of PKD2^{gutΔΔ} mice (Fig 5C) and animals treated with CRT0066101 inhibitor (Fig 6F).

“The gut microbiota analysis in Pkd2gutΔ/Δ mice is interesting. However, it is not clear whether this microbiota remodeling is a cause or consequence of Pkd2 gene deficiency and/or improvement of glucose homeostasis. This should be discussed.”

We agree with reviewer #2 that this is a very interesting aspect of our study. Therefore we have partially moved these panels to the main figures. We have also discussed these results in the appropriate section (Page 19).

“The figure 6 nicely shows that blocking Pkd2 activity with CRT0066101 improves glucose homeostasis, decrease TG absorption, and can ameliorate glucose parameter after obesity is established. Although this is an interesting approach, its is not clear whether this pharmacological strategy is directly targeting intestinal Pkd2. The authors should demonstrate that the inhibitor effects are mediated though the inhibition of intestinal Pkd2. In addition, is there any effects of this inhibitor on insulin secretion during glucose tolerance test?”

To test this we have measured the activity of PKD2 and all PKDs in the intestine, liver, and adipose tissue. Of note, PKD2 activation was efficiently inhibited by CRT0066101 in the intestine, but not in the other organs. Similarly, other PKDs were not inhibited in the liver and adipose tissue (Fig 6C and EV5A-B).

Also insulin levels were decreased in mice treated with PKD inhibitor when compared control animals (Fig EV5F).

“Minor : there are some typos, errors in the references, and errors in figure labelling (fig 4C->D).”

We have fixed it.

Referee #3 (Remarks for Author):

Previous studies have shown that Protein Kinase D1 (PKD1) promotes obesity by inhibiting energy dissipation in adipocytes and PKD3 increases hepatic insulin resistance. The current study by Jonathan T. et al seeks to investigate an unexpected role of PKD2 in lipid homeostasis. The major findings of this study include: 1) that PKD2 enhances chylomicron-mediated TG transfer in enterocytes; 2) PKD2 increases chylomicron size to enhance the TG secretion on the basolateral side of the mouse and human enterocytes; 3) PKD2 phosphorylated APOA4 that is one of the apolipoproteins associated with chylomicrons; 4) Deletion, inactivation of PKD2 improved HFD- induced obesity, diabetes and improves the gut microbiota profile in mice; 5) Pharmacological inhibition of PKD2 improved HFD-induced obesity, diabetes and changed gut microbiota profile in mice. Therefore, the authors demonstrated that PKD2 is a key signaling molecule affecting dietary fat absorption and may serve as a potential target for treatment of obesity. It is an interesting study, however, there are several issues to be addressed.

We thank reviewer #3 for a careful assessment of our manuscript.

“Specific Comments:”

“1. The gender information of the mice used in the experiments should be included in both methods and figure legends.”

We have included this information in the method section and figures legends.

“2. Previous studies have found that protein kinase D1 deletion in adipocytes enhances energy dissipation and protects against adiposity. Does CRT 0066101 affect PKD1 activity?”

CRT 0066101 inhibits all PKDs. However, in our study, we have used a relatively low dose of the inhibitor which was given orally. This was sufficient to inhibit PKDs in the intestine, but not in the liver or adipose tissue (Fig 6C and EV5A-B). We have also discussed this issue in the appropriate section of the discussion.

“3. In Figure EV1, Pkd2 ki/ki mice have smaller adipocyte sizes of EpiWAT and SubWAT compared to control group, does deletion of Pkd2 affect the browning of white adipose tissue or affect the mitochondrial biogenesis, or increase UCP1 expression?”

We have measured the expression of major beige adipocyte markers in subcutaneous adipose tissue isolated from PKDki/ki mice. We have observed that, except *Slc6a8* which was downregulated in the mice without active PKD2, there were no significant changes in the

expression of *Ucp1*, *Cidea*, *Bmp7*, *Prdm16*, *Ppara*, *Pgc1a* among other beiging markers (Fig EV1A)

“4. Previous studies found that inhibition of PKD might lead to increased accumulation of lipids and fibrosis in the liver. It would be important to determine the levels of TG in livers of Pkd2 ki/ki mice and pharmacological inactivation of Pkd2 in mice.”

We have determined the hepatic content of TG in both mouse models. PKD2 inactivation resulted in a significant reduction of TG content in the liver, however pharmacological inhibition of PKD showed no significant differences (Fig 1F and EV5H).

“5. CRT treatment reduced the weight of BAT in Figure 6F, does CRT treatment affect the histological features of BAT?”

Treatment of mice with CRT 0066101 reduced HFD-induced hypertrophy and favored a healthier multilocular BAT (Fig EV5I).

“6. It is shown that Pkd2 ki/ki mice exhibited similar levels of ALT and AST, does CRT treatment affect the levels of AST and ALT?”

CRT 0066101 did not alter AST and ALT levels in mice (Fig EV5G).

“7. “we analyzed specific changes in Bacterial operational taxonomic units (OTUs) in the different gut segments. In Duodenum, we identified members of Bacteroides, which are associated with weight loss (Turnbaugh et al., 2006), to be only present in Pkd2 gutΔ/Δ mice, but completely absent in control animals”. It might be important to move these data from the Supplemental Material into the main text.”

Following the suggestion of reviewer #3, we have presented these data on the main figures.

“8. Figure 5F, why is there no error bar for this graph? Also please include the labels for the y-axis.”

We have included it.

“9. In the Discussion, the authors discuss potential therapies of inhibition of PKD2 and inhibition of PKD-dependent signaling in the intestine as a potential targeted therapeutic strategy for obesity and T2D, do T2D or obese patients have increased levels or activities of PKD2?”

We have analyzed the activity of PKD2 in the intestine of obese patients who underwent Roux-en-Y gastric bypass surgery. Of note, the levels of PKD2 activity (phosphorylation on S876) correlated positively with the levels of triglycerides in circulation and the percentage of glycated hemoglobin (Fig 7F-H). We also found a negative correlation with HDL levels in those patients (which did not reach significance) (Fig 7I).

“Minor:

Typo: "ThermoFischer" in Page 37.”

Corrected

9th Mar 2021

Dear Dr. Sumara,

Thank you for the submission of your revised manuscript to EMBO Molecular Medicine. We have now received the enclosed reports from the three referees who re-reviewed your manuscript. As you will see, they are now supportive of publication, and I am therefore pleased to inform you that we will be able to accept your manuscript, once the following minor points will be addressed:

1/ Main manuscript text:

- Please answer/correct the changes suggested by our data editors in the main manuscript file (in track changes mode). This file will be sent to you in the next couple of days. Please use this file for any further modification.

- Please provide up to 5 keywords.

- In the references, please make sure that you only list 10 authors before et al.

- Please move the Material and Methods section after the discussion.

- Material and methods:

- o Cells: please indicate the origin of the cells, and whether they were tested for mycoplasma contamination.

- o Antibodies: please indicate the concentrations used in your study.

- o Human samples: please include a statement confirming that the experiments conformed to the principles set out in the WMA Declaration of Helsinki and the Department of Health and Human Services Belmont report.

- Statistics: Please indicate in the figures or in the legends the exact n= and exact p= values along with the statistical test used. You may provide these values as a supplemental table in an Appendix file (an Appendix file should contain a table of content).

- Please include a data availability section: primary datasets produced in this study need to be deposited in an appropriate public database (see <https://www.embopress.org/page/journal/17574684/authorguide#dataavailability>). If no new dataset was generated, please indicate: "This study includes no data deposited in external repositories"

2) Figures and tables:

- The reference for Fig. EV2K in the main text is missing, please add a callout for this figure panel.

- Please upload the two tables as separate files and rename them Table EV1 and Table EV2.

3) Checklist: please provide more details in sections D/8-9 (animal models) and sections E/11-12. In section F/18-19, you indicated "done", however I could not find the Data Availability section in your manuscript (please also see comment above). Please adjust accordingly.

4) Thank you for providing The Paper Explained. Please include it in the main manuscript file.

5) Thank you for providing a nice synopsis image and the accompanying text. I slightly edited it to fit our style and format, please let me know if you agree with the following:

'We show that upon fat ingestion, Protein Kinase D2 stimulates chylomicron-mediated triglyceride absorption in the intestine. Targeting PKD2, genetically or with small molecule inhibitors, reduces triglycerides absorption and prevents the development of obesity in mice and presumably in

humans.

- PKD2 enhances chylomicron size and therefore chylomicron-mediated triglycerides absorption.
- PKD2 phosphorylates chylomicron-associated lipoprotein, APOA4.
- Inhibition of PKD2 diminishes obesity and associated diabetes.
- PKD2 activity correlates with triglycerides levels in obese patients.'

6) As part of the EMBO Publications transparent editorial process initiative (see our Editorial at <http://embomolmed.embopress.org/content/2/9/329>), EMBO Molecular Medicine will publish online a Review Process File (RPF) to accompany accepted manuscripts.

This file will be published in conjunction with your paper and will include the anonymous referee reports, your point-by-point response and all pertinent correspondence relating to the manuscript. Let us know whether you agree with the publication of the RPF and as here, if you want to remove or not any figures from it prior to publication.

I look forward to receiving your revised manuscript.

Yours sincerely,

Lise Roth

Lise Roth, PhD
Editor
EMBO Molecular Medicine

To submit your manuscript, please follow this link:

Link Not Available

***** Reviewer's comments *****

Referee #1 (Remarks for Author):

All my points have been addressed sufficiently.

Referee #2 (Remarks for Author):

The authors have satisfactorily replied to my concerns, including several new data/informations. The link between microbiota and PKD2 remains an open question that deserves to be studied in the near future.

Referee #3 (Comments on Novelty/Model System for Author):

This is an important study with novel results demonstrating that PKD2 represents a key signaling node promoting dietary fat absorption and may serve as an attractive target for treatment of obesity. The authors have addressed all of the comments and the manuscript has been improved.

Referee #3 (Remarks for Author):

The authors have adequately addressed the concerns and the manuscript has been improved. Nice work.

The authors performed the requested editorial changes.

17th Mar 2021

Dear Dr. Sumara,

Thank you for sending the revised files. I have looked at everything, and all is fine. I am therefore very pleased to accept your manuscript for publication in EMBO Molecular Medicine!

We note that the link provided in the Data Availability section does not yet link to accessible data, please make sure that the dataset is public before acceptance of the manuscript.

Your manuscript will be sent to our publisher to be included in the next available issue of EMBO Molecular Medicine.

Please read below for additional important information regarding your article, its publication and the production process.

Congratulations on a nice study!

Yours sincerely,

Lise Roth

Lise Roth, Ph.D
Editor
EMBO Molecular Medicine

Follow us on Twitter @EmboMolMed
Sign up for eTOCs at embopress.org/alertsfeeds

Corresponding Author Name: Grzegorz Sumara
Journal Submitted to: EMBO Molecular Medicine
Manuscript Number: EMM-2020-13548